# Inductive Quantum Embedding

**Santosh K. Srivastava**∗**, Dinesh Khandelwal**∗ **, Dhiraj Madan**∗**, Dinesh Garg**∗**,**
**Hima Karanam, L Venkata Subramaniam**
IBM Research AI, India
sasriva5, dhikhand1, dmadan07, garg.dinesh, hkaranam, lvsubram@in.ibm.com

## Abstract

Quantum logic inspired embedding (aka Quantum Embedding (QE)) of a Knowledge-Base (KB) was proposed recently by Garg et al. [1]. It is claimed that the QE preserves the logical structure of the input KB given in the form of unary and binary predicates hierarchy. Such structure preservation allows one to perform Boolean logic style deductive reasoning directly over these embedding vectors. The original QE idea, however, is limited to the transductive (not inductive) setting. Moreover, the original QE scheme runs quite slow on real applications involving millions of entities. This paper alleviates both of these key limitations. We start by reformulating the original QE problem to allow for the induction. On the way, we also underscore some interesting analytic and geometric properties of the solution and leverage them to design a *faster* training scheme. As an application, we show that one can achieve state-of-the-art performance on the well-known NLP task of *fine-grained entity type classification* by using the inductive QE approach. Our training runs 9-times faster than the original QE scheme on this task.

## 1 Introduction

*Knowledge Representation (KR)* is a field of AI aiming at representing the worldly information inside a computer so as to solve complex tasks in an automated manner. *Automated Reasoning* is another important field of AI and goes hand in hand with KR because one of the main purposes of explicitly representing knowledge is to be able to reason about that knowledge, make inferences, assert new knowledge, etc. Virtually all knowledge representation techniques have an automated reasoning engine as part of the overall system, for example, automated question answering, document search, automated dialogue systems, etc.

Predominantly, there are two approaches to KR - (i) *Discrete symbolic representation*, (ii) *Continuous vector representation* [2]. In symbolic representation, knowledge facts are represented by symbols, and some form of logical reasoning (for example, first-order logic) is used to infer new facts and make deductions. Symbolic reasoning is *exact* but *slow, brittle,* and *noise-sensitive*. Popular examples of symbolic form of knowledge representation include *semantic nets, frames,* and *ontologies* [3].

Vector form representation is a complementary approach to the symbolic form representation. Vector form representation stems from the field of Statistical Relational Learning [4, 5], where knowledge is embedded into a vector space using a distributional representation capturing (dis)similarities among entities and predicates. Vector representations are *fast, noise-robust*, but *approximate*. Despite being fast, most of the vector representation techniques do not offer any explicit means of preserving the input KB's logical structure inside the vector space. Thus, the logical reasoning tasks typically experience a lower accuracy when working with vector representation than the symbolic representation of the knowledge. Making these observations, Dominic [6, 7] suggested using Quantum Logic [8] style framework for word meaning disambiguation problem in keyword-based search engines. This

---

∗Equal contribution.

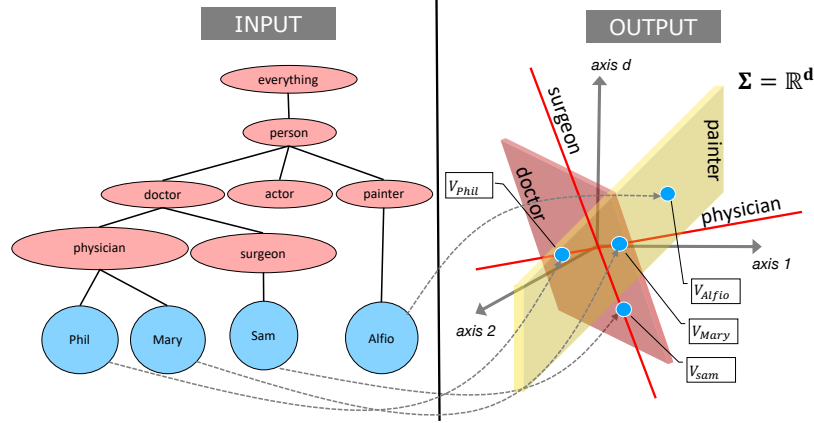

Figure 1: Illustration of quantum embedding of unary predicate hierarchy. Image inspiration from [2]

idea was further baked by Garg et al. [1], and an approach called *Quantum Embedding (QE)* was proposed, which is claimed to preserve the logical structure of the given KB. In what follows, we recapitulate the idea behind QE followed by research gaps and our contributions.

**A Recapitulation of Quantum Embedding:** To recap the original idea of QE proposed in [1], we have drawn Figure 1. The left side in Figure 1 depicts a typical unary predicate hierarchy, where red oval nodes denote unary predicates (aka concepts), and blue circular nodes denote entities. The right side depicts a cartoon illustration of the QE of the KB given on the left side. The key aspects of QE follow from this illustrative diagram. 1) QE maps an *entity i* to a *vector $x_i$*, and a *unary (or binary) concept predicate $C_j$* to a *linear subspace $S_j$*, respectively, inside a given $d$-dimensional vector space $\mathbb{R}^d$ (complex space $\mathbb{C}^d$ in the more general case of the binary predicate). 2) The above mapping is designed so that for each entity and predicate, its logical relationship with other entities and predicates remains intact in the embedding space $\mathbb{R}^d$. Specifically, QE satisfies the following design desiderata. First, each entity vector $x_i$ lies within the subspace $S_j \subset \mathbb{R}^d$ corresponding to the concept $C_j$ to which entity $i$ belongs (as per left side KB). Second, for any pair of concepts $(C_j, C_k)$ in the left side KB, if $C_k$ is a parent of $C_j$, the corresponding subspaces $S_j, S_k$ satisfy containment relationship $S_j \subseteq S_k$. Third, the subspace corresponding to $(C_j \texttt{ AND } C_k)$ and $(C_j \texttt{ OR } C_k)$ are given by $S_j \cap S_k$ and $S_j + S_k$, respectively, where $+$ means vector sum and not set-theoretic union. Fourth, the logical $\texttt{NOT}$ of concept $C_j$ is given by $S_j$'s orthogonal complement, and denoted by $S_j^\perp$. Now, for any given pair $(x_i, C_j)$, the probability that entity $i$ belongs to the concept $C_j$ is proportional to the squared length of the orthogonal projection of vector $x_i$ onto the subspace corresponding to $C_j$, and this comes from the *principle of measurement* in *Quantum Mechanics* [8]. Finally, it is important to recall that unlike Boolean logic, *distributive law* does not hold in the quantum logic. That means, although we have $C_i \texttt{ AND } (C_j \texttt{ OR } C_k)$ being equal to $(C_i \texttt{ AND } C_j) \texttt{ OR } (C_i \texttt{ AND} C_k)$, the corresponding subspaces need not have $S_i + (S_j \cup S_k)$ being equal to $(S_i + S_j) \cup (S_i \cup S_k)$. As suggested in [1], this problem can be alleviated by restricting to axis-parallel subspaces; we also leverage this fact in our formulation. For binary predicates, the QE [1] idea works almost the same except that one needs to operate in complex space $\mathbb{C}^d$ and with pairs of entities, where an entity pair $(i, j)$ is represented via $(x_i + \iota x_j) \in \mathbb{C}^d$.

The method proposed in [1] takes the left side structure of Figure 1 as the input and outputs a structure of the right side. However, when we receives a new entity (that is, a new blue node) belonging to some existing concept class at a later point in time, the original method [1] can not compute its embedding in an incremental manner. Instead, the original method needs to be rerun from the scratch by including this new entity.

**Research Gaps, Motivation, and Contributions:** We observed the following gaps in the original QE proposal [1]. 1) QE is *transductive* and not *inductive*. Given an unseen test entity, there is no prescribed way to compute its QE incrementally. 2) QE expresses each of its design desiderata using an appropriate loss function and then combines all these losses into a non-convex objective function whose solution gives the QE. The original paper suggests using stochastic gradient descent (SGD) scheme [9] to find QE. However, it is extremely slow on practical problems that typically involve millions of entities in a hundred-dimensional space, resulting in more than a hundred million variables. 3) The original paper hardly sheds any light on the resulting embedding's analytic/geometric properties.

However, we observed that many such useful properties could be derived - both *theoretically* and *empirically*. These insights indeed guided our way towards developing a faster scheme to solve the original problem. Motivated by these gaps and observations, in this paper,

**[Sections 2]** We propose a reformulation of the original model [1] that allows the ingestion of entities' initial feature vectors and thereby, opening a way for the inductive extension. We call this optimization problem as an **I**nductive **Qu**antum **E**mbedding (IQE) problem. Our reformulation is a non-convex, integer-valued, and constrained optimization problem.

**[Sections 3]** We propose a custom-designed *alternating optimization* scheme for solving the IQE problem. This scheme is 9-times faster on a certain application as compared to SGD. In both Sections 2 and 3, we also discuss the analytic/geometric properties of an optimal solution of the IQE problem.

**[Sections 4-5]** Lastly, we consider an important Natural Language Processing (NLP) task, namely *Fine-grained Entity Type Classification (FgETC)*. We show that IQE formulation can be made inductive, and one can infer the QE of unknown test entities for this task. We also show that one can achieve state-of-the-art performance on this task by using our inductive QE approach. Moreover, IQE trains 9-times faster than the original QE approach for this task.

## 2 Quantum Embedding for Inductive Setting

In this section, we develop a reformulation of the QE problem [1] for unary predicates. Extension to binary predicate is possible (discussed later). We use algebraic properties of the orthogonal projection matrices to build such a reformulation (this section) and its solution (next section). For proofs, please refer to Section 1 of the supplementary material.

Let the given KB contain $n$ entities (blue nodes in Figure 1) and $m$ leaf-level concepts. Corresponding to each leaf concept $C_j$, let $S_j$ be the subspace in the embedding space $\mathbb{R}^d$. Let $\mathcal{F} = \{\mathbf{P}_1, \ldots \mathbf{P}_m\}$ be a set of orthogonal projection matrices onto the subspaces $S_1, \ldots, S_m$, respectively. Let $x_i$ be a vector representation of the entity $i$ and $\mathbb{1}_j(i)$ be an indicator function expressing whether an entity $i$ belongs to concept $C_j$ or not. Typically, an entity vector $x_i$ associated with concept $C_j$ would not be lying perfectly in the subspace $S_j$. As suggested in [1], the amount of such imperfection can be measured by the squared Euclidean distance between the point $x_i$ and its orthogonal projection onto the subspace $S_j$. That is, $\sum_{i=1}^{n} \sum_{j=1}^{m} \|x_i - \mathbf{P}_j x_i\|^2 \mathbb{1}_j(i)$, where the sum is taken over all subspaces and entities. Note, all $\mathbf{P}_j$s being equal to the identity matrix would trivially minimize this loss function. To avoid this, we include an additional *negative sampling loss* term that tries to decrease the projection length of an entity $x_i$ onto all the concept spaces except the ones to which it belongs. That is, if it is not mentioned explicitly in the input KB that $x_i$ belongs to the concept $C_j$, we force it to be closer to the orthogonal complement subspace $S_j^\perp$. The loss function, therefore, becomes:

$$g(x, \mathbf{P}) \overset{\text{def}}{=} \sum_{i=1}^{n} \sum_{j=1}^{m} \|x_i - \mathbf{P}_j x_i\|^2 \mathbb{1}_j(i) + \lambda \|x_i - \mathbf{Q}_j x_i\|^2 \bar{\mathbb{1}}_j(i), \tag{1}$$

where $(x, \mathbf{P}) = (x_1, \ldots, x_n, \mathbf{P}_1, \ldots, \mathbf{P}_m)$, $\bar{\mathbb{1}}_j(i) = (1 - \mathbb{1}_j(i))$, and $\mathbf{Q}_j = (\mathbf{I} - \mathbf{P}_j)$ is an orthogonal projection matrix for the orthogonal complement subspace $S_j^\perp$. Here, $\lambda$ is the tuning parameter of the negative sampling: $\lambda = 0$ subscribes to the Open World Assumption (OWA), and as $\lambda$ increases, the formulation moves closer to the Closed World Assumption (CWA) [3].

Next, to inculcate *inductive* behavior in our formulation, we assume some prior information is available for each training/test entity in the form of a *feature vector* $f_i \in \mathbb{R}^p$. $f_i$ could be a word embedding, for example. We now bias an entity's QE vector $x_i$ to correlates with its feature vector $f_i$. We leverage such a correlation to infer the QE of an unseen test entity from its feature vector. We appeal to a simple linear model for inducing such a bias and minimize the following regression loss $\sum_{i=1}^{n} \|x_i - \mathbf{W} f_i\|^2$, where $\mathbf{W} \in \mathbb{R}^{d \times p}$ becomes model parameters. Adding this to (1), the loss function becomes

$$g(x, \mathbf{W}, \mathbf{P}) \overset{def}{=} \sum_{i=1}^{n} \sum_{j=1}^{m} \|\mathbf{Q}_j x_i\|^2 \mathbb{1}_j(i) + \lambda \|\mathbf{P}_j x_i\|^2 \bar{\mathbb{1}}_j(i) + \alpha \sum_{i=1}^{n} \|x_i - \mathbf{W} f_i\|^2 \tag{2}$$

where $\alpha$ is the hyperparameter. Note that, $x_i = 0$ could trivially drive the first two terms of (2) to zero. To avoid such a trivial solution, we add a constraint, $\|x_i\|^2 = 1$. Before we move, we would like to mention that although we use a linear model $\mathbf{W}$ to induce bias, we don't use the same model

at the inference time because we found its capacity to be quite low in our experiments. Therefore, we train a separate deep-net based higher-capacity model $\Phi(\cdot)$ that maps feature vector $f_i$ into the biased QE $x_i$ (obtained through the above formulation). We refrain from using $\Phi(\cdot)$ instead of $\mathbf{W}$ in the above formulation because of two reasons. 1) Mathematical analysis and closed-form solution become intractable, 2) $\Phi(\cdot)$ tends to dominate other QE loss terms and overfits $x_i$'s onto $f_i$'s.

Distributive law is one of the important properties of classical logic. Garg et al. [1] showed that a sufficient condition for the distributive law to hold is that the projection matrices satisfy the *commutative property*. For the same reason, we also include the constraint that the projection matrices $\mathbf{P}_j \in \mathcal{F}$ must satisfy the following commutative property: $\mathbf{P}_i \mathbf{P}_j = \mathbf{P}_j \mathbf{P}_i$ for all $\mathbf{P}_i, \mathbf{P}_j \in \mathcal{F}$. We also include a regularization term to avoid degenerate solution where all $\mathbf{P}_j$s are the same. For this, we force each leaf subspace $S_j$ to be as orthogonal to the other leaf subspace $S_k$ as possible. In other words, $\mathbf{P}_j$ is orthogonal to $\mathbf{P}_{j+1}, \ldots \mathbf{P}_m$ for $j = 1, \ldots, m$. Given that we are forcing commutative property between projection matrices, using the Corollary 2 of Section 1 of the supplementary material, orthogonality of $\mathbf{P}_j$ to $\mathbf{P}_{j+1}, \ldots \mathbf{P}_m$ can be modeled by regularization term, namely $\sum_{j'>j}^{m} \operatorname{tr}(\mathbf{P}_j \mathbf{P}_{j'})$ for each $\mathbf{P}_j$. The overall loss function, therefore, now becomes

$$f(x, \mathbf{W}, \mathbf{P}) \stackrel{def}{=} g(x, \mathbf{W}, \mathbf{P}) + \sum_{j=1}^{m} \sum_{j'>j}^{m} \gamma \operatorname{tr}(\mathbf{P}_j \mathbf{P}_{j'}). \tag{3}$$

The $\gamma$ is a control coefficient that measures the degree of orthogonality between a pair of subspaces. These orthogonality terms try to drive the solution towards low-rank projections ($\mathbf{P}_j = 0$ in the worst case). To avoid such a degenerate solution, we put the constraint saying the rank of the orthogonal projection matrices $\mathbf{P}_j$ in the optimal solution must be greater than equal to $r$. We found experimentally that we obtain a better solution with this extension. The overall optimization problem then becomes as follows

$$\begin{aligned}
\text{Minimize} \quad & f(x, \mathbf{W}, \mathbf{P}) \\
\text{subject to} \quad & \|x_i\|^2 = 1, \quad i = 1, \ldots, n, \tag{4} \\
& \mathbf{P}_i \mathbf{P}_j = \mathbf{P}_j \mathbf{P}_i \quad \text{for all} \quad \mathbf{P}_i, \mathbf{P}_j \in \mathcal{F} \tag{5} \\
& \operatorname{tr}(\mathbf{P}_j) \geq r, \text{ where } r \geq 1, \ \mathbf{P}_j \in \mathcal{F}, \tag{6}
\end{aligned}$$

where, the objective is given by (3) and variables are $x_1, x_2, \ldots, x_n, \mathbf{W}$, and $\mathbf{P}_1, \mathbf{P}_2, \ldots, \mathbf{P}_m$. The problem data are indicator functions $\mathbb{1}_j(i)$ and feature vectors $f_i$ in (2). The hyper-parameters are $\lambda, \alpha, \gamma,$ and $r$ defined in (1), (2), (3), and (6), respectively. The above optimization problem is non-convex. We call this problem as **I**nductive **Qu**antum **E**mbedding (IQE) problem. In what follows, we state three key properties of the IQE problem. 1) Rotational invariance, 2) Probabilistic interpretation of entities' memberships, and 3) NP-Hardness.

**Theorem 1. Rotational Invariance of IQE:** *The IQE problem is invariant to rotational transformation. That is, if $\{x_1, \ldots, x_n, W, P_1, \ldots P_m\}$ is a solution of the IQE problem, and $V$ is any $d \times d$ orthonormal matrix then $\{Vx_1, \ldots, Vx_n, VW, VP_1 V^T \ldots VP_m V^T\}$ is also a solution.*

**Proof:** The proof is given in the Section 2 of the supplementary material.

**Probabilistic Interpretation of Entities' Memberships:** For any given pair $(x_i, \mathbf{P}_j)$, the probability that entity $i$ belongs to the concept $C_j$ is proportional to the squared length of orthogonal projection of vector $x_i$ onto $\mathbf{P}_j$, i.e. $\|\mathbf{P}_j x_i\|^2$. It comes from the principle of measurement in *Quantum Mechanics* [8]. Our problem can now be interpreted as maximizing the probability of an entity belonging to the ground truth concepts and minimizing the probability of entity belonging to the other concepts.

**NP-Hardness:** We observed that if we use different $\gamma$ for each $(j, j')$ pair, the IQE problem becomes NP-hard. The proof is given in the Section 3 of the supplementary material.

**Binary Predicate Extension:** The above framework can be extended to a binary predicate setting. For binary concept predicate $C_j$, we are given entity pairs that are related through predicate $C_j$. For example, `Mary` *is_mother_of* `Sam`. Like [1], we can denote the binary predicate *"is_mother_of"* via a subspace of complex space $\mathbb{C}^d$, `Mary` (`Sam`) via vectors $x_{\texttt{Marry}}$ ($x_{\texttt{Sam}}$) $\in \mathbb{R}^d$, and the pair (`Marry`, `Sam`) via $x_{\texttt{Marry}} + \iota x_{\texttt{Sam}} \in \mathbb{C}^d$. Section 4 of the Supplementary Material sheds further light on this extension.

# 3 Solution

We solve the IQE problem using alternating minimization. First, we clamp the variables $\mathbf{W}, \mathbf{P}_1, \mathbf{P}_2, \ldots \mathbf{P}_m$ as well as constraints involving them and solve the resulting IQE problem over variables $x_1, x_2, \ldots, x_n$. In the second step, we clamp $x_1, x_2, \ldots, x_n$ and $\mathbf{P}_1, \mathbf{P}_2, \ldots \mathbf{P}_m$ as well as their constraints (4) - (6) and solve the resulting IQE problem over $\mathbf{W}$. Finally, in the third step, we clamp $x_1, x_2, \ldots, x_n$ and $\mathbf{W}$ as well as their constraints (4) and solve for $\mathbf{P}_1, \mathbf{P}_2, \ldots \mathbf{P}_m$. We repeat these three alternating steps till convergence. The steps are given in Algorithm 1.

---

**Algorithm 1:** Alternating Minimization Scheme for IQE Problem

---

Pick appropriate values for the hyperparameters $d, \lambda, \alpha, \gamma, r, p$ ;
Given a KB, construct the indicator function $\mathbb{1}_j(i)$, feature vectors $f_i$, and initialize $x_i$ randomly;
**while** *(Solution do not converge)* **do**
    Clamp variables $(\mathbf{W}, \mathbf{P})$ and solve the problem given by (7) - (8) **[called as Problem 1]**;
    Clamp variables $(x, \mathbf{P})$ and solve the problem over $\mathbf{W}$ given by (9) **[called as Problem 2]**;
    Clamp variables $(x, \mathbf{W})$ and solve the IQE problem over $\mathbf{P}_j$s **[called as Problem 3]**;

---

**Problem 1 (Optimizing over $x$):** Observe, when $\mathbf{W}, \mathbf{P}_1, \mathbf{P}_2, \ldots \mathbf{P}_m$ are clamped to the values that satisfy constraints (5) and (6), the objective function (3) is convex quadratic in $x_i$'s. The resulting problem becomes Quadratically Constrained Quadratic Program (QCQP) and separable in the variables $x_1, \ldots, x_n$. Therefore, we can solve this QCQP problem by solving a separate problem (called *Problem 1*) for each $x_i$. Ignoring the constant term and denoting $\mathbf{I}_d$ as a $d$-by-$d$ identity matrix, the Problem 1 for an $x_i$ is

$$\text{Minimize} \quad x_i^T \mathbf{R}_i x_i - 2x_i^T c_i, \tag{7}$$

$$\text{subject to} \quad \|x_i\|^2 = 1, \text{ where } \mathbf{R}_i = \alpha \mathbf{I}_d + \sum_{j=1}^m \mathbf{Q}_j \mathbb{1}_j(i) + \lambda \mathbf{P}_j \bar{\mathbb{1}}_j(i) \text{ and } c_i = \alpha \mathbf{W} f_i. \tag{8}$$

We show in Section 5 of the supplementary material that the optimal solution of (7-8) is given by $(\mathbf{R}_i - \mu \mathbf{I}_d) x_i = c_i$, where Lagrange multiplier $\mu$ (for equality constraint) can be obtained by solving the following secular equation [10]: $\sum_{j=1}^d c_{ij}^2 / (\lambda_j - \mu)^2 = 1$ and $\mu < \lambda_1$, $c_{ij}$ is the $j^{th}$ component of the vector $c_i$, and $\lambda_1 \le \lambda_2 \ldots \le \lambda_d$ are the eigenvalues of $\mathbf{R}_i$. The LHS of this secular equation is a monotonically increasing function of $\mu$, taking value in the range of $(0, +\infty)$, as we move $\mu$ in the interval $(-\infty, \lambda_1)$. Therefore, it must have one unique solution in the interval $(-\infty, \lambda_1)$. We obtained $\mu$ numerically using the bisection method [11].

**Problem 2 (Optimizing over $\mathbf{W}$):** We consider the problem of finding an optimal solution $\mathbf{W} \in \mathbb{R}^{d \times p}$, when $(x, \mathbf{P})$ are clamped to the values that satisfy constraints (4)-(6). The objective function (3) as a function of $\mathbf{W}$, ignoring constant terms, reduces to

$$f(\mathbf{W}) \quad = \quad \sum_{i=1}^n \text{tr}\left(\mathbf{W}^T \mathbf{W} f_i f_i^T - 2\mathbf{W} f_i x_i^T\right) = \text{tr}\left(\mathbf{W}\mathbf{F}\mathbf{F}^T \mathbf{W}^T\right) - 2\text{tr}\left(\mathbf{X}\mathbf{F}^T \mathbf{W}^T\right), \quad (9)$$

where $\mathbf{F} \in \mathbb{R}^{p \times n}$ and $\mathbf{X} \in \mathbb{R}^{d \times n}$ are matrices whose columns are $f_i$ and $x_i$ respectively. In (9), we use the property that the trace is invariant under cyclic permutation and transpose. Computing the gradient of $f(\mathbf{W})$ with respect to $\mathbf{W}$ and setting it to zero, gives

$$(\mathbf{W}\mathbf{F} - \mathbf{X})\mathbf{F}^T \quad = \quad 0 \quad \implies \quad \mathbf{W} = \mathbf{X}\mathbf{F}^\dagger. \tag{10}$$

where $\mathbf{F}^\dagger \in \mathbb{R}^{n \times p}$ is the pseudo-inverse of $\mathbf{F}$.

**Problem 3 (Optimizing over $\mathbf{P}$):** Here, we consider optimizing over the subspaces when $x_1, \ldots, x_n$, and $\mathbf{W}$ are clamped to their current estimates. Since all the projection matrices $\mathbf{P}_j$'s commute, they are simultaneously diagonalizable via a common orthogonal matrix (due to Theorem 9 given in Section 1 of the supplementary material). Furthermore, Projection matrices can be viewed in terms of diagonal matrices because IQE is rotationally invariant. Taking $\mathbf{P}_j = \text{diag}(y_{j,1}, ..., y_{j,d})$ where each $y_{j,k} \in \{0, 1\}$, the loss function (3) can be written in the following manner:

$$\sum_{k=1}^d \sum_{j=1}^m \left( \sum_{i \in S_j} x_{i,k}^2 + y_{j,k} \left( \lambda \sum_{i \notin S_j} x_{i,k}^2 - \sum_{i \in S_j} x_{i,k}^2 \right) + \sum_{j' > j}^m \gamma y_{j,k} y_{j',k} \right)$$

where the first term inside the sum is constant and can be dropped. The coefficient of $y_{j,k}$ is denoted by

$$\phi_{j,k} \quad \overset{def}{=} \quad \lambda \sum_{i \notin S_j} x_{i,k}^2 - \sum_{i \in S_j} x_{i,k}^2. \tag{11}$$

For the reason discussed later, we refer to $\phi_{j,k}$ as *potential function*. The Problem 3 now becomes,

$$\text{minimize} \quad \sum_{k=1}^{d} \left( \sum_{j=1}^{m} y_{j,k} \phi_{j,k} + \gamma \sum_{j=1}^{m} \sum_{j'>j}^{m} y_{j,k} y_{j',k} \right), \tag{12}$$

subject to $y_{j,k} \in \{0,1\}$ and rank constraint $\sum_{k=1}^{d} y_{j,k} \geq r$. We solve the problem (12) approximately via a heuristic. We first minimize (12) without considering the rank constraint. Subsequently, we add some of the $y_{j,k}$'s greedily to fulfill the rank constraint and increase the objective value as minimal as possible. Note, in the absence of rank constraint, the objective function (12) is separable in $k$. Therefore, for each $k$, minimization over $y_{1,k}, y_{2,k}, \ldots, y_{m,k}$ reduces to

$$\min_{n_k} \sum_{j=1}^{n_k} \tilde{\phi}_{j,k} + \gamma \binom{n_k}{2}, \tag{13}$$

where $n_k = |\{j : y_{j,k} = 1\}|$ and $\tilde{\phi}_{1,k} \leq \tilde{\phi}_{2,k} \ldots \leq \tilde{\phi}_{m,k}$ is the sorting of $\phi_{j,k}$'s in increasing order. We made two simplifications to reach objective (13): 1) If $n_k = t$ is the solution, then $y_{j,k} = 1$ corresponding to the smallest $t$ values of $\phi_{j,k}$, and the rest are zeros. 2) Only non zero $y_{j,k}, y_{j',k}$ pairs contribute to the sum of the orthogonality terms. The second term in (13) is precisely the number of ways to choose 2 entries of $y_{j,k}$ amongst those taking the value 1. Suppose we have chosen $(\ell - 1)$ smallest entries of $\phi_{j,k}$, then, an additional contribution of adding the $\ell^{th}$ smallest entry to the solution is $\tilde{\phi}_{\ell,k} + \gamma \left( \binom{\ell}{2} - \binom{\ell-1}{2} \right) = \tilde{\phi}_{\ell,k} + \gamma(\ell - 1)$. This increment also increases for each successive $\ell$. From these observations, we see that it suffices to sort all the $\phi_{j,k}$'s in increasing order and then greedily keep assigning $y_{j,k} = 1$ until the objective function value decreases. Then, we greedily add some of the $y_{j,k}$'s to fulfil the rank constraint and with an increase in the objective value as minimal as possible. In Section 6 of the supplementary material, we provide a pseudo-code for the above procedure and an alternative way to solve (12).

**Geometry of Entity Vectors and Concept Subspaces:** Recall, IQE problem outputs axis-parallel subspaces. That is, any concept $C_j$ is denoted by an axis parallel subspace $S_j$ of the quantum embedding space. Also, for an entity $i$, $\Pr(x_i \in C_j) \propto \|\mathbf{P}_j x_i\|^2$. Therefore, if $C_j$ has an axis $\ell$ in its subspace, as square of $\ell^{th}$ coordinate of $x_i$ increases, $\Pr(x_i \in C_j)$ also increases. In lieu of this fact, our proposed scheme for Problem 3 has an interesting geometric interpretation. Recall, our scheme (in Problem 3) to find the best set of axes for the subspace $\mathbf{P}_j$ is based on the idea of reducing the potential function (11), which takes the difference of two terms. The first term in $\phi_{j,k}$ is the sum of the square of component $k$ across all those entity vectors which do not belong to the subspace $S_j$. On the other hand, the second term is the sum of the square of component $k$ across all those entity vectors that belong to the subspace $S_j$. The constant $\lambda$ is a parameter that sets the relative importance of the two terms. Thus, if the potential function $\phi_{j,k}$ is least across all the dimensions, the axis $k$ is a critical axis in the representation of the subspace $S_j$.

## 4   Fine-Grained Entity Type Classification

*Named Entity Recognition (NER)* is a basic NLP task. It comprises two subtasks - (i) detecting mentions of named entities in the input text, (ii) classifying the identified mentions into a predefined set of type classes (aka schema). The classical NER literature [12, 13, 14, 15] focuses on *coarse-grained entity type classification* comprising only a few types in the schema, for example, *person, location, organization,* etc. The recent literature [16, 17, 18, 19, 20, 21, 22] has shifted the focus towards *fine-grained entity type classification (FgETC)*, where schema contains more than 100 type classes arranged in a hierarchy. The FgETC task critically depends on the context. For example, consider two sentences - (i) **Obama** *was born in Honolulu, Hawaii.*, (ii) **Obama** *taught constitutional law at the University of Chicago Law School for twelve years.* One can infer *person* as the only type of **Obama** from the first sentence, whereas it can be refined to the *lawyer* in the second one. In FgETC, entities mentions are already given, and the task is to classifying these mentions into a given fine-grained type hierarchy. A state-of-the-art technique for FgETC [23] uses an attention-based neural network to capture an entity's context and uses the same to classify.

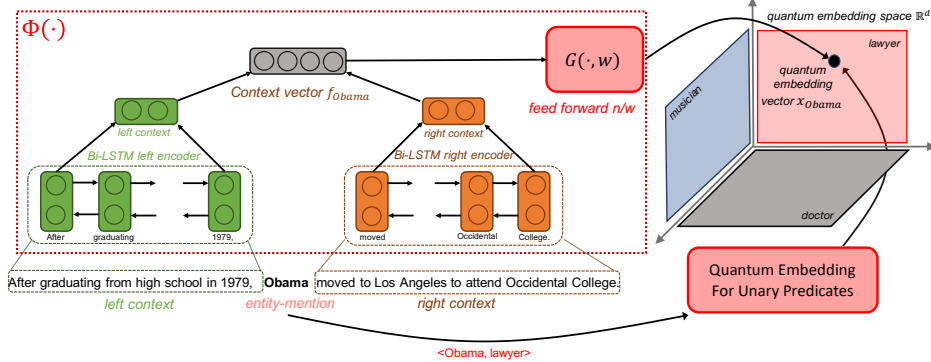

Figure 2: Architecture for learning feature map $\Phi(\cdot)$ for FgETC task.

This section aims to demonstrate that QE learned by our proposed IQE method are quite effective in training a *task-agnostic* map $x_{\text{test}} = \Phi(\mathbf{f}_{\text{test}})$, which maps the set of context tokens' feature vectors $\mathbf{f}_{\text{test}}$ of an unseen entity to its QE vector $x_{\text{test}}$. The inferred quantum embedding can then be used as a high-quality feature vector in downstream tasks. We use FgETC as an example of a downstream task and show that the class labels inferred from $\Phi(\mathbf{f}_{\text{test}})$ achieve state-of-the-art performance. For comparison sake, we also train the map $\Phi(\cdot)$ using the QE learned from the original scheme of Garg et al. [1], which does not make use of initial feature vectors $f_i$. The map $\Phi(\cdot)$ offers the following advantages: 1) By definition, a QE vector comprises an intuitive geometric interpretation regarding classes in the hierarchy to which it belongs. Hence, no separate classifier needs to be trained. 2) It alleviates the transductive limitation of QE and makes it inductive. 3) The classification accuracy is on par with the state-of-the-art. The details of the fitting map $\Phi(\cdot)$ are given next.

For each training entity $i$, we take its left (right) context (including the entity $i$) and tokenize the same. Each token is replaced with its 300 dimensional Glove vector [24]. The set of these Glove vectors is denoted by $\mathbf{f}_i$. Adding the left (right) context vectors from this set $\mathbf{f}_i$ gives us the left (right) context vector for the entity $i$ and we denote this by $f_{il}(f_{ir})$. Appending (or adding) left and right context vectors $f_{il}$ and $f_{ir}$ gives us the overall context vector $f_i$ for the entity $i$ having size 600 (or 300). Using $f_i$ and the given type classes, we generate quantum embedding $x_i$ for each entity $i$ in the training set (via Algorithm 1). Next, we learn the feature map $\Phi(\cdot)$ between the set $\mathbf{f}_i$ of left-right context token vectors and the quantum embedding $x_i$ of an entity $i$. We model this $\Phi(\cdot)$ via a Bi-LSTM network followed by a feed-forward neural network $G(\cdot, \mathbf{w})$, where $\mathbf{w}$ is its parameters (see Figure 2). $G(\cdot, \mathbf{w})$ consists of fully connected layers with a $\tanh$ activation function. The output dimension for each of these layers is $d$, output of the last layer is normalized to the unit norm, and the loss is *Euclidean distance*. The strategy to predict test labels is as follows. We trace the type hierarchy in a top-down manner. We pick type $j$ as the entity's class if we have already picked its parent type, say $j'$, as its class, and we have $((\|P_{j'}\Phi(\mathbf{f}_{\text{test}})\|^2 - \|P_j\Phi(\mathbf{f}_{\text{test}})\|^2)/\|P_{j'}\Phi(\mathbf{f}_{\text{test}})\|^2) < \tau/\delta$, where $\tau, \delta$ are hyper-parameters and $\delta = 1$ if $j'$ has less than 10 children, otherwise it is tuned. At root, we pick the type $j'$ that has the highest $\|P_{j'}\Phi(\mathbf{f}_{\text{test}})\|^2$ score. Here, $\mathbf{f}_{\text{test}}$ is the set of left-right context vectors.

## 5 Experiments

**Dataset:** For FgETC task, the relevant datasets include `FIGER` [16, 19], `TypeNet` [21], `Ontonote` [13, 25]. We experiment with the `FIGER` dataset. This dataset consists of 127 different entity types arranged in two levels of the hierarchy (106 leaf and 21 internal nodes). The detailed hierarchy is shown in the supplementary material (Section 7). Key statistics about this dataset is given in Table 4.

**Quantum Embedding Step:** As discussed earlier, we first generate the quantum embedding for each training entity using Algorithm 1. We run this algorithm until convergence, which was less than 10 iterations (see Figure 6 for the convergence trend). Various hyper-parameter values pertaining to this step are summarized in Table 2. The last column captures the values tried during parameter selection. The second last column gives the final chosen value. The choice was made through training accuracy.

To test the quality of generated quantum embeddings, we randomly sampled 20 entities from a randomly chosen 10 classes and plotted their quantum embeddings $x_i$ via a 2-dimensional t-SNE plot [26]. We repeated the same process but replacing $x_i$ with context feature vector $f_i$. These plots are shown in Figures 3 and 4, respectively. It is clear from these plots that

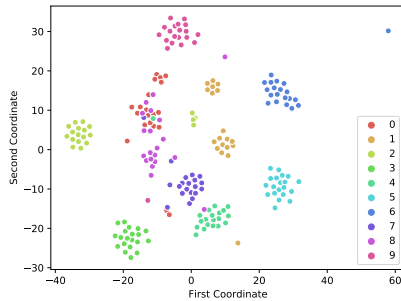

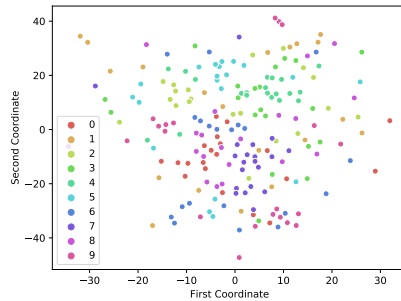

Figure 3: t-SNE plot for QE obtained via IQE

Figure 4: t-SNE plot for Glove embeddings

quantum embeddings are much better clustered in terms of class labels. We further did one more analysis. For every pair $(i, j)$ of these randomly selected entities, we computed pairwise distance between them in the following manner: $d_{qe} = \|x_i - x_j\|^2$; $d_{fv} = \|f_i - f_j\|^2$; $d_{tree} = $ *Path length in the hierarchy tree between class labels of entities i and j.* We plotted a linear regression plot between $d_{tree}$ and $d_{qe}$ (as well as $d_{fv}$), as shown in Figure 5. It is clear from this figure that quantum embeddings denote better separation behavior about class labels.

Finally, we also made the clock-time comparison of our alternating scheme with the original SGD based scheme [1] on the FIGER dataset. This comparison is shown in Table 1. In this table, $t_{qe}$ and $i_{qe}$ denote *per iteration time* and *number of iterations*, respectively, taken during any QE method training. The quantities $t_{nn}$ and $i_{nn}$ denote *average time per epoch* and *number of epochs*, respectively, for training the feature map $\Phi(\cdot)$. The average time $t_{nn}$ is approximately the same for both the methods. There are now two speedup factors - one without including $T_{nn}$ and other with including $T_{nn}$. Note, the neural network $\Phi(\cdot)$ comes after QE in our pipeline (as shown in Figure 2), and it learns the mapping from the input sentence feature vector to the QE, irrespective of which method was used to generate the QE (our method or the original method [1]). Therefore, a more meaningful comparison would be to compare our method with the original method [1] only in terms of the time taken to generate QE (i.e. $T_{qe}$). As per this comparison, our method is 9.08 times faster than the original method [1].

| Method | $t_{qe}$ | $i_{qe}$ | $T_{qe} = t_{qe} \times i_{qe}$ | $t_{nn}$ | $i_{nn}$ | $T_{nn} = t_{nn} \times i_{nn}$ | $T_{qe} + T_{nn}$ |
|--------|---------|---------|------------------------------|---------|---------|------------------------------|-------------------|
| Ours   | 510.6   | 6       | 3063.6                       | $\approx 1275$ | 6 | 7650 | 10713.6 |
| [1]    | 27.8    | 1000    | 27800.0                      | $\approx 1275$ | 6 | 7650 | 35450.0 |
| Speedup |        |         | **9.07**                     |         |         |                              | **3.31** |

Table 1: Time comparison of our method with original method of [1]. All times are in *seconds*.

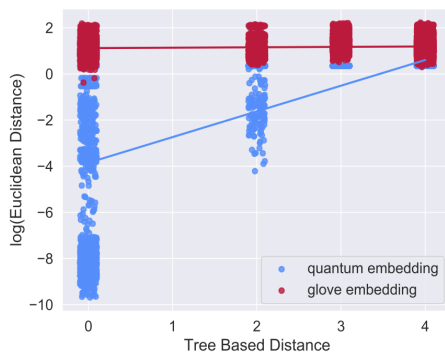

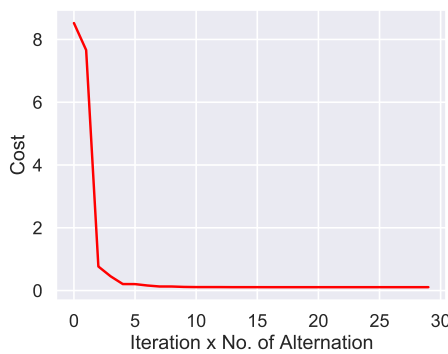

Figure 5: Correlation between entities distances

Figure 6: Convergence behavior of Algorithm 1

**Learning to Transform Context Vectors into Quantum Embedding Vectors:** In this step, we learn the map $\Phi(\cdot)$ between left-right context vectors of a training entity and its quantum embedding

(achieved in the previous step). Various hyper-parameters used in this step are given in Table 3. We implemented the IQE model using PyTorch[2].

| Parameter | Symbol | Value | Tuning-Range |
|---|---|---|---|
| QE dimension | $d$ | 300 | $[100, 300]$ |
| $f_i$ dimension | $p$ | 300 | $[300, 600]$ |
| Reg. parameter | $\lambda$ | 10 | $[1, 2, 5, 10, 100]$ |
| Reg. parameter | $\alpha$ | 0.001 | $[0.0001–0.1]$ |
| Reg. parameter | $\gamma$ | 1 | $[0.001–1]$ |
| Minimum Rank | $r$ | 5 | $[1, 2, 5]$ |

Table 2: Hyperparameters for learning QE

| Parameter | Symbol | Value | Tuning-Range |
|---|---|---|---|
| Glove vectors dim | | 300 | $[300]$ |
| Learning rate | $\eta$ | 0.0001 | $[0.0001–0.01]$ |
| Threshold Parameter | $\tau$ | 0.15 | $[0.1–0.95]$ |
| Threshold Parameter | $\delta$ | 5 | $[1–15]$ |
| # Bi-LSTM layers | | 1 | $[1,2]$ |
| # Layers in $G(\cdot, \cdot)$ | | 3 | $[1–4]$ |

Table 3: Hyperparameters for learning $\Phi(\cdot)$

**Results:** In Table 5, we have summarized the performance of our quantum embedding based scheme for the FgETC task on the test set of the `FIGER` dataset. We have used the performance metrics that are standard in this literature [16]. We observe that we are able to beat most of the baselines on this task and obtain state-of-the-art results. With regard to the strongest baseline of [23], we are up by 4-points on *Accuracy* but down by 2.5-points on $F1$. At this moment, we would like to highlight that our inductive model for QE is trained in a task agnostic manner. The trained model for inductive quantum embedding can be used for any downstream task, for example, the FgETC task in this case. We also evaluated the performance (row 1 in Table 5) if we train $\Phi(\cdot)$ using the original QE.

| Split | #Sentences | #Entities | #Tokens |
|---|---|---|---|
| Train | 1505765 | 2690286 | 1342679 |
| Test | 434 | 564 | 713602 |

Table 4: Summary of `FIGER` dataset. We used 10% of the *Test* set as development set to tune parameters $\tau, \delta$, and rest 90% for the final evaluation.

| Method | Accuracy | Macro $F_1$ | Micro $F_1$ |
|---|---|---|---|
| QE (Garg et al.[1]) | 0.383 | 0.420 | 0.347 |
| FIGER (Liang et al. [16]) | 0.471 | 0.617 | 0.597 |
| AFET (Ren et al. [19]) | 0.533 | 0.693 | 0.664 |
| Attentive[†] (Shimaoka et al. [23]) | 0.589 | **0.779** | **0.749** |
| **IQE (Ours)** | **0.631** | 0.764 | 0.724 |

Table 5: Performance on test set. [†] They used $95\% - 5\%$ split of the *Train* set for training and development.

**Insights:** i) QE obtained through IQE formulation is an improved representation over the Glove context vector (as evident from Figures 3, 4, and 5). ii) One can extend the QE framework for the inductive setting in a task agnostic manner. Also, the induction quality is far superior when QE is learned through IQE formulation than the original formulation [1] (as evident from Table 5). iii) Our proposed solution for the IQE problem converges 9.08-times faster than the original scheme of [1].

# 6   Conclusions and Future Directions

This paper offers two critical improvements to the recent idea of QE [1] - from transductive to the inductive setting, and a faster scheme to compute QE. Both the improvements were demonstrated through a well-known NLP task of FgETC. Our proposed IQE approach achieves state-of-the-art performance on this task and runs 9-times faster than the original QE scheme. An important future direction would be to provide a richer model to ingest initial feature vectors into the IQE model. This may include pair-wise information, Bayesian prior, attention-based prior, etc.

# 7 Broader Impact

Knowledge Representation (KR) is an important subfield of Artificial Intelligence and plays a crucial role in designing any complex AI system that aims to mimic human like reasoning. Prominent examples of such a system include *automated question answering, document search, and retrieval, product recommendation, automated dialogue/conversation, automated navigation,* etc. The purpose of KR is to encode a symbolic Knowledge-Base (KB) within a machine reasoning system. These KBs could be domain/application-specific, such as *medical, fashion, retail, e-commerce,* etc; could be in *public domain*, or *proprietary to an organization/enterprise*. The most common examples of KBs in the public domain include DBPedia [27] and WordNet [28].

There are two key approaches to KR - (i) *Discrete symbolic representation*, (ii) *Continuous vector representation* [2]. In symbolic representation, knowledge facts are represented by symbols, and some form of logical reasoning (for example, first-order logic) is used to infer new facts and make deductions. Symbolic reasoning is *exact* but *slow, brittle,* and *noise-sensitive*. Vector representation stems from the field of Statistical Relational Learning [4, 5], where knowledge is embedded into a vector space using a distributional representation capturing the (dis)similarities among entities and predicates. Vector representations are *fast, noise-robust*, but *approximate*. Despite being fast, most of the vector representation techniques do not offer explicit means of preserving the logical structure of the input Knowledge-Base (KB) inside vector space. Making these observations, Dominic [6, 7] hinted at using Quantum Logic [8] framework to fix the word meaning disambiguation problem in keyword-based search engines. This idea was further developed and extended recently by Garg et al. [1], where they have proposed a new approach for vector representation of symbolic KBs – called as *Quantum Embedding (QE)*.

This paper identifies two critical gaps in the QE approach [1] for KR. The end outcome is a refinement of the QE idea bridging these gaps. Specifically, we noticed that the original idea of quantum embedding [1] is *transductive* (and *not inductive*) in nature. That is, it can learn to embed a given symbolic KB; however, for an unseen knowledge element (entity or predicate), it does not prescribe any recipe to embed the same in an incremental way. The only way seems to restart from scratch and reproduce the whole embedding by including the new knowledge element. This, in our view, limits the applicability of the quantum embeddings in practical applications. Further, we also noticed that the computational scheme suggested in [1] for generating quantum embedding is quite slow because it is based on the general-purpose Stochastic Gradient Descent (SGD) algorithm.

To address the above gaps, we first propose a reformulation of the original model [1] that allows ingestion of entities' initial feature vectors and thereby, opening a way for the inductive extension. We call this optimization problem as **I**nductive **Qu**antum **E**mbedding (IQE) problem. Next, we discover some interesting analytic and geometric properties and leverage them to design a *faster* training scheme. As an application, we consider the well-known NLP task of *fine-grained entity type classification* [16, 17, 18, 19, 20, 21, 22]. We show that one can use IQE formulation for this task to infer quantum embeddings of unseen test entities and subsequently use those quantum embedding (instead of initial feature vectors) to infer the entity's class label. We show that our proposed IQE approach achieves a state-of-the-art performance on this task and runs 9-times faster than the original QE scheme.

Although, a good part of this paper is theoretical in nature, the refinements proposed in this paper can impact the adoption of QE idea for knowledge representation in broad range of applications including *automated question answering, document search and retrieval, product recommendation, automated dialogue/conversation systems,* etc. These applications are now an integral part of our daily lives. We witness them in *customer support service, e-commerce platforms, online education platforms, voice-based search, home automation, vehicle navigation,* etc. Improving such systems' performance offers huge societal benefits such as cost/time savings, removing repetitive tasks, and increasing autonomy for the elderly/children. However, this also poses societal risks, including *adversarial attacks, hacking into such systems, and biasing them with malicious intents, risk of having different kinds of biases in training data,* etc. We would encourage the research community to study further the extent to which such representations can be manipulated by an adversary either by biasing the training data or hacking the system.

## 8   Funding Disclosure and Competing Interests

The authors state that no funding was associated in direct support of this work and neither any of the authors receive third party funding or third party support for this work. The authors declare no competing financial interests.

**Acknowledgments**

We would like to thank Shajith Ikbal for helping us with running the approach given in [1]. We would also like to thank Sanjeeb Dash, Kush Varshney, and Dennis Wei for their feedback on an early draft of this paper. Finally, our sincere thanks to all the anonymous reviewers for their valuable comments and suggestions to improve the manuscript.

## Footnotes

[2]The code to train and evaluate our model is available at https://github.com/IBM/e2r/tree/master/neurips2020.

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
