[Supplementary Material]

# Inductive Quantum Embedding
# (Supplementary Material)

**Santosh K. Srivastava**∗, **Dinesh Khandelwal**∗ , **Dhiraj Madan**∗, **Dinesh Garg**∗,
**Hima Karanam, L Venkata Subramaniam**
IBM Research AI, India
sasriva5, dhikhand1, dmadan07, garg.dinesh, hkaranam, lvsubram@in.ibm.com

## 1 Algebra of Subspace

A theorem involving a subspace can be interpreted as theorem about orthogonal projection matrix in the sense that subspace may be expressed as a linear transformation on $\mathbb{R}^d$ [1]. Therefore we formulate the IQE problem in term of orthogonal projection matrices. To formulate the IQE problem as a non-convex optimization problem, we need some facts about orthogonal projection matrices. In this section, we review some important concepts and key results about orthogonal projection matrices. We have sketched out the proofs for some of the theorems, while the proof for other theorems can be found in [2], [3], [4].

**Theorem 1. Intersection of subspaces [4]:** *Let $\boldsymbol{P}_1$ and $\boldsymbol{P}_2$ be the orthogonal projectors onto the subspaces $S_1$ and $S_2$ respectively. In general, the subspaces $S_1$ and $S_2$ are not necessary disjoint. The necessary and sufficient condition for the matrix $\boldsymbol{P}_1\boldsymbol{P}_2$ to be an orthogonal projector onto the subspace $S_1 \cap S_2$ is*

$$\boldsymbol{P}_1\boldsymbol{P}_2 = \boldsymbol{P}_2\boldsymbol{P}_1. \tag{1}$$

**Proof:** We are giving the proof here for the sake of self sufficiency.

Assume $\mathbf{P}_1\mathbf{P}_2 = \mathbf{P}_2\mathbf{P}_1$. We show that $\mathbf{P}_1\mathbf{P}_2$ is an orthogonal projector onto the subspace $S_1 \cap S_2$.

$$\begin{aligned}
\left(\mathbf{P}_1\mathbf{P}_2\right)^2 &= \mathbf{P}_1\left(\mathbf{P}_2\mathbf{P}_1\right)\mathbf{P}_2 = \mathbf{P}_1\left(\mathbf{P}_1\mathbf{P}_2\right)\mathbf{P}_2 \\
&= \mathbf{P}_1^2\mathbf{P}_2^2 = \mathbf{P}_1\mathbf{P}_2
\end{aligned}$$

this establishes that $\mathbf{P}_1\mathbf{P}_2$ is a projection matrix. To show it is orthogonal projection, consider $\left(\mathbf{P}_1\mathbf{P}_2\right)^T$, which simplifies to

$$\left(\mathbf{P}_1\mathbf{P}_2\right)^T = \mathbf{P}_2^T\mathbf{P}_1^T = \mathbf{P}_2\mathbf{P}_1 = \mathbf{P}_1\mathbf{P}_2,$$

therefore $\mathbf{P}_1\mathbf{P}_2$ is an orthogonal projection matrix. To show that it is indeed an orthogonal projection matrix onto the subspace $S_1 \cap S_2$, let $\mathbf{x} \in S_1 \cap S_2$. Then, $\mathbf{P}_1\left(\mathbf{P}_2 x\right) = \mathbf{P}_1 x = x$. Furthermore, let $x \in (S_1 \cap S_2)^\perp = S_1^\perp + S_2^\perp$ and $x = x_1 + x_2$, where $x_1 \in S_1^\perp$ and $x_2 \in S_2^\perp$. Then,

$$\begin{aligned}
\mathbf{P}_1\mathbf{P}_2 x &= \mathbf{P}_1\mathbf{P}_2 x_1 + \mathbf{P}_1\mathbf{P}_2 x_2 \\
&\overset{(a)}{=} \mathbf{P}_2\mathbf{P}_1 x_1 + \mathbf{0} \\
&= \mathbf{0}
\end{aligned}$$

where, we used the assumption $\mathbf{P}_1\mathbf{P}_2 = \mathbf{P}_2\mathbf{P}_1$ to derive the first term in $(a)$, while the second term is zero because $x_2 \in S_2^\perp$ and hence it must be that $\mathbf{P}_2 x_2 = \mathbf{0}$. This proves the fact that $\mathbf{P}_1\mathbf{P}_2$ is an orthogonal projection matrix onto the subspace $S_1 \cap S_2$. To prove the converse, assume that $\mathbf{P}_1\mathbf{P}_2$ is an orthogonal projection onto the subspace $S_1 \cap S_2$. It is easy to see $\mathbf{P}_1\mathbf{P}_2 = \left(\mathbf{P}_1\mathbf{P}_2\right)^T = \mathbf{P}_2^T\mathbf{P}_1^T = \mathbf{P}_2\mathbf{P}_1$.
□

---

∗Equal contribution.

**Corollary 2.** *Let $P_1$ and $P_2$ be the orthogonal projectors onto the subspaces $S_1$ and $S_2$ respectively. The necessary and sufficient condition that $S_1$ and $S_2$ are orthogonal subspaces if and only if*

$$P_1P_2 = P_2P_1 = O, \tag{2}$$

*where $O$ is a zero matrix.*

**Theorem 3. Inclusion of subspace:** *Let $P_1$ and $P_2$ be the orthogonal projectors onto the subspaces $S_1$ and $S_2$ respectively. The following statements are equivalent:*

1. $S_1 \subset S_2$.

2. $P_2P_1 = P_1$.

3. $P_1P_2 = P_1$.

**Proof:**
**[1 $\implies$ 2]:** Assume $S_1 \subset S_2$. For every $x \in \mathbb{R}^d$, $P_1 x \in S_1 \subset S_2$. Therefore $P_2(P_1 x) = P_1 x$, which implies $P_2 P_1 = P_1$.

**[1 $\implies$ 3]:** Since $S_1 \subset S_2$, this implies orthogonal complement subspaces $S_1^\perp$ and $S_2^\perp$ satisfies $S_2^\perp \subset S_1^\perp$. Apply the above proof to the orthogonal complement projectors $Q_1, Q_2$ gives

$$\begin{aligned}
Q_1 Q_2 &= Q_2 \\
(I - P_1)(I - P_2) &= I - P_2 \\
P_1 P_2 &= P_1.
\end{aligned}$$

**[2 $\implies$ 1]:** Assume $P_2 P_1 = P_1$. For every $x \in \mathbb{R}^d$, $P_1 x \in S_1$, which implies $P_1 x = P_2 P_1 x \in S_2$, which implies $S_1 \subset S_2$.

**[3 $\implies$ 1]:** Assume $P_1 P_2 = P_1$, this implies $Q_1 Q_2 = Q_2$, which in turn implies $S_2^\perp \subset S_1^\perp$, which implies $S_1 \subset S_2$. $\qquad\square$

**Theorem 4. Union of subspaces [2]:** *Let $P_1$ and $P_2$ be the orthogonal projectors onto the subspaces $S_1$ and $S_2$ respectively, and let $P_{1+2}$ denote the orthogonal projector onto the subspace $S_{1+2} = S_1 + S_2$. Then the following statements are equivalent:*

1. $P_1 P_2 = P_2 P_1$.

2. $P_{1+2} = P_1 + P_2 - P_1 P_2$.

**Corollary 5.** *Let $P$ denote the orthogonal projection onto the subspace $S = S_1 + S_2$, and let $P_1, P_2$ be the orthogonal projectors onto the subspaces $S_1$ and $S_2$ respectively. If $S_1$ and $S_2$ are orthogonal, then*

$$P = P_1 + P_2.$$

**Theorem 6. De Morgan's law of subspaces:** *Let $P_1$ and $P_2$ be the orthogonal projectors onto the subspaces $S_1$ and $S_2$, respectively. If $P_1$ commutates with $P_2$, then*

$$(S_1 \cap S_2)^\perp = S_1^\perp + S_2^\perp.$$

**Proof:** Assume $(S_1 \cap S_2)^\perp$. According to the theorem 1, this implies

$$\begin{aligned}
I - P_1 P_2 &= I - (I - Q_1)(I - Q_2) \\
&= I - (I - Q_1 - Q_2 + Q_1 Q_2) \\
&= Q_1 + Q_2 - Q_1 Q_2,
\end{aligned}$$

which is equivalent to $S_1^\perp + S_2^\perp$. $\qquad\square$

**Theorem 7. Distributive law of subspaces [4]:** *Let $P_i, P_j, P_k$ denote the orthogonal projector onto the subspace $S_i, S_j, S_k$ respectively. If $P_i P_j = P_j P_i$, $P_j P_k = P_k P_j$, and $P_i P_k = P_k P_i$, then the following relations of distributive law of subspaces hold:*

$$\begin{aligned}
S_i + (S_j \cap S_k) &= (S_i + S_j) \cap (S_i + S_k), \\
S_j + (S_i \cap S_k) &= (S_i + S_j) \cap (S_j + S_k), \\
S_k + (S_i \cap S_j) &= (S_i + S_k) \cap (S_j + S_k). \tag{3}
\end{aligned}$$

Commutativity of the orthogonal projection matrices is an important condition for the distributive law of subspaces to hold. One way to satisfy this condition is to consider the following theorem.

**Theorem 8. Simultaneous Diagonalization [3]:** *Let $\mathcal{F}$ be a set of orthogonal projection matrices. Projection matrices satisfy a pairwise commutative property $\boldsymbol{P}_i \boldsymbol{P}_j = \boldsymbol{P}_j \boldsymbol{P}_i$ for all $\boldsymbol{P}_i, \boldsymbol{P}_j \in \mathcal{F}$ if and only if there exist a common orthogonal matrix $\boldsymbol{V}$ such that*

$$\boldsymbol{P}_j = \boldsymbol{V} \boldsymbol{D}_j \boldsymbol{V}^T \quad \text{for all} \quad \boldsymbol{P}_j \in \mathcal{F}, \tag{4}$$

*where $\boldsymbol{D}_j$ is a $d \times d$ diagonal matrix with 0 and 1 on the diagonal.*

The purpose of the theorem 8 is threefold. First, it enables distributive property of subspaces to hold true through (4). Second, it implies set $\mathcal{F}$ is finite. Under the condition of the theorem 8, $\mathbf{V}$ is fixed for each $\mathbf{P}_j \in \mathcal{F}$, (4) implies that $\mathbf{P}_j$ is isomorphic to the diagonal matrix $\mathbf{D}_j$. Since the diagonal of the diagonal matrix $\mathbf{D}_j$ is a $d$-component binary vector, there are $2^d$ orthogonal projection matrices possible in a $d$ dimensional Euclidean space $\mathbb{R}^d$. Third, when $\mathbf{V}$ equals to identity matrix, $\mathbf{P}_j$ equals to $\mathbf{D}_j$ which is an axis-parallel subspace.

**Axis-Parallel Subspace:** Axis-parallel subspace is a subspace whose boundaries are either parallel or perpendicular to the standard basis. In a $d$ dimensional Euclidean space $\mathbb{R}^d$, axis-parallel subspace is spanned by the subset of the standard basis vectors $\{e_1, e_2, \ldots, e_d\}$. Given a $d$ dimensional Euclidean space $\mathbb{R}^d$, there are $2^d$ distinct axis-parallel subspaces possible. The following theorem captures the essence that projection matrices onto the axis-parallel subspace are diagonal matrices.

**Theorem 9. [4]:** *Let $\boldsymbol{A} = [e_{i_1} | e_{i_2} | \ldots | e_{i_m}]$, where $e_{i_1}, e_{i_2}, \ldots, e_{i_m}$ is subset of a standard basis vectors of $\mathbb{R}^d$. Then the orthonormal projector $\boldsymbol{P}$ onto the subspace $S = \mathrm{range}(A)$ spanned by the basis vectors $e_{i_1}, e_{i_2}, \ldots, e_{i_m}$ is given by*

$$\boldsymbol{P} = \boldsymbol{A} \boldsymbol{A}^T = \sum_{j=1}^{m} e_{i_j} e_{i_j}^T, \tag{5}$$

*which is a $d \times d$ diagonal matrix with 0 and 1 along the diagonal.*

## 2 Rotational Invariance of IQE

We show that the (IQE) problem, defined in the Section 2 of the main paper, is invariant to rotational transformation.

**Proof for Theorem 1 of the Main Paper**
In the objective (3) of the main paper, if we replace each $x_i$, $\mathbf{W}$, and $\mathbf{P}_j$ by $\mathbf{V}x_i$, $\mathbf{V}\mathbf{W}$, and $\mathbf{V}\mathbf{P}_j\mathbf{V}^T$ respectively, where $\mathbf{V}$ is a $d$-by-$d$ orthonormal matrix, then it becomes

$$\sum_{i=1}^{n} \sum_{j=1}^{m} \left( \left\| \mathbf{V}\mathbf{Q}_j \left( \mathbf{V}^T \mathbf{V} \right) x_i \right\|^2 \mathbb{1}_j(i) + \lambda \left\| \mathbf{V}\mathbf{P}_j \left( \mathbf{V}^T \mathbf{V} \right) x_i \right\|^2 \bar{\mathbb{1}}_j(i) \right) +$$

$$+ \gamma \sum_{j=1}^{m} \sum_{j'>j} \mathrm{tr} \left( \mathbf{V}\mathbf{P}_j \left( \mathbf{V}^T \mathbf{V} \right) \mathbf{P}_{j'} \mathbf{V}^T \right) + \alpha \sum_{i=1}^{n} \left\| \mathbf{V}x_i - \mathbf{V}\mathbf{W}f_i \right\|^2$$

$$= \sum_{i=1}^{n} \sum_{j=1}^{m} \left( \left\| \mathbf{V}\mathbf{Q}_j x_i \right\|^2 \mathbb{1}_j(i) + \lambda \left\| \mathbf{V}\mathbf{P}_j x_i \right\|^2 \bar{\mathbb{1}}_j(i) \right) + \gamma \sum_{j=1}^{m} \sum_{j'>j} \mathrm{tr} \left( \mathbf{V}\mathbf{P}_j \mathbf{P}_{j'} \mathbf{V}^T \right) +$$

$$+ \alpha \sum_{i=1}^{n} \left\| \mathbf{V} \left( x_i - \mathbf{W}f_i \right) \right\|^2 \tag{6}$$

$$= \sum_{i=1}^{n} \sum_{j=1}^{m} \left( \left\| \mathbf{Q}_j x_i \right\|^2 \mathbb{1}_j(i) + \lambda \left\| \mathbf{P}_j x_i \right\|^2 \bar{\mathbb{1}}_j(i) \right) + \gamma \sum_{j=1}^{m} \sum_{j'>j} \mathrm{tr} \left( \mathbf{P}_j \mathbf{P}_{j'} \right) + \alpha \sum_{i=1}^{n} \left\| x_i - \mathbf{W}f_i \right\|^2 \tag{7}$$

In (6) we used $\mathbf{V}^T\mathbf{V} = \mathbf{I}$, since $\mathbf{V}$ is an orthonormal matrix. In (7) we used the facts that the 2-norm of the vector and trace of the matrix are both invariant to orthogonal transformation [5]. Therefore, the objective function is invariant to rotational transformation. Similarly, it could be

shown that the constraints (4)-(6) of the IQE problem, described in the main paper, are also rotational invariant. Therefore if $\{x_1, \ldots, x_n, \mathbf{W}, \mathbf{P}_1, \ldots \mathbf{P}_m\}$ is the solution of the IQE, then $\{\mathbf{V}x_1, \ldots, \mathbf{V}x_n, \mathbf{VW}, \mathbf{VP}_1\mathbf{V}^T, \ldots \mathbf{VP}_m\mathbf{V}^T\}$ will also be the solution. Thus, IQE is rotational invariant. $\qquad\square$

## 3 NP-Hardness of IQE Problem

Here, we consider an extension of the objective function covered in the main paper, wherein we allow the coefficient of orthogonality penalty term to take multiple values. This makes the problem NP-hard as we show below. Formally, we consider the objective function where the coefficient for orthogonality terms depends on both $j$ and $j'$. We keep it as $\gamma_{j,j'}$. We consider the following problem:

$$\text{minimize} \quad \sum_{k=1}^{d}\left(\sum_{j=1}^{m}\theta_k + y_{j,k}\phi_{j,k} + \sum_{j\neq j'}\gamma_{j,j'}y_{j,k}y_{j',k}\right), \tag{8}$$

$$\text{such that} \quad \sum_{k=1}^{d}y_{j,k} \geq r \;\; \text{and} \;\; y_{j,k} \in \{0,1\},$$

$$\text{where} \quad \phi_{j,k} \overset{def}{=} \lambda\sum_{i\notin S_j}x_{i,k}^2 - \sum_{i\in S_j}x_{i,k}^2. \tag{9}$$

**Independent Sets in Graphs:** Let $G = (V, E)$ denote a graph of $|V|$ nodes. An independent set (aka stable set) $S$ in $G$ is a subset of the vertices of $G$ such that for every two vertices in $S$, there is no edge connecting the two. The *independent set* problem is the problem of finding a independent set with highest cardinality in a given graph. Finding an independent set of largest size is a classical NP-hard problem, with many diverse applications [6, 7, 8].

**Theorem 10.** *NP-Hardness: The optimization problem (8) is NP-hard even for the simplest case of $d = 1$, no rank constraint ($r = 0$), and $\gamma$ taking only 2 possible values depending on $j, j'$.*

**Proof:** We can prove the above claim by reducing the independent set problem to this problem.

We keep the dimension $d = 1$, so there is no summation over $k$. We set number of subspaces to the number of vertices $|V|$ in the input problem. We set $\phi_j = -\frac{\delta}{2.|V|} \;\forall j$ for some small $\delta$. We also set

$$\gamma_{j,j'} = \begin{cases} 0 & if(j,j') \notin S \\ \delta & if(j,j') \in S \end{cases}$$

Now, this problem is equivalent to selecting a maximum subset of vertices such that there no edge between the selected vertices. The optimal value of this optimization problem is same as the optimal value of the given instance of stable set problem. $\qquad\square$

## 4 Binary Predicate Extension

In the case of binary predicates, we are typically given a set of entities, a set of binary predicates, and a set of relation triples in the form of $(e_s, r, e_o)$ as training examples, where entities $e_s, e_o$ are known as subject and object, respectively for the triple and $r$ denotes a relation between them. For example, Mary is_mother_of Sam. In this example, $e_s = $ Mary, $e_o = $ Sam, $r = $ is_mother_of.

To proceed further, we make following notional convention.

$$\begin{aligned} n &= \text{Number of unique entities} \\ m &= \text{Number of unique binary relations} \\ t &= \text{Number of relation triples given in training data} \\ E &= \text{Set of entities. This means, } |E| = n \\ R &= \text{Set of binary relations. This means, } |R| = m \\ T &= \text{Set of training triples. This means, } |T| = t \end{aligned}$$

We make a few more conventions as follows.

1. We denote any triple given in the training set by $(e_i, r_k, e_j)$ where $i, j \in [n]$, $k \in [m]$, $e_i, e_j \in E$, and $r_k \in R$.

2. Observe, we must always have $T \subseteq (E \times R \times E)$. Thus, we can define the training set in an alternate manner by defining an indicator function $\mathbb{1}_k(ij)$ as follows.

$$\mathbb{1}_k(ij) = \begin{cases} 1 & \text{if } (e_i, r_k, e_j) \in T \\ 0 & \text{o/w} \end{cases} ; \forall e_i, e_j \in E, r_k \in T$$

where $i, j = 1 \to n$, $k = 1 \to m$. Now, we write the overall IQE formulation for the binary predicate. We make the following assumptions for this formulation.

1. To simplify the discussion here and remain focused on binary predicates extension, we ignore the term related to ingestion of initial feature vectors $f_i, f_j$ for the entities $e_i, e_j$; because it can be done in a manner similar to what we did for the unary predicate case in the main paper.

2. There may be a hierarchy among binary relations as well. For the modeling sake, we assume that it is a flat hierarchy, i.e. all the binary relations are connected to root in the relation hierarchy and each of these binary relations have one or more instances given as training examples. Taking care of multi-level is quite straightforward by incorporating corresponding loss terms as suggested in the section on *Recapitulation of Quantum Embedding* in the main paper.

3. Unlike our unary predicate model in the main paper and also unlike the assumption made by [4] for the binary predicate case, we do not adhere to the assumption of axis-parallel concept spaces and instead admits non axis-parallel concept subspaces. While the axis-parallel concept spaces assumption buys the distributive law holding true (as shown by [4]), this assumption severely limits the representation capability of the quantum embedding for binary predicates case because the cross-correlation terms between subject and object entities gets canceled in the original formulation. The net result being that the resulting representation admit large number of invalid triples into a relation subspace. This effect can be seen in the poor performance of the original quantum embedding [4] on WN18 dataset. Furthermore, for the simple link prediction kind of tasks (such as the ones required in WB18 and FB15K datasets), the test queries are much simpler and we don't require distributive law to hold true in general. We believe this is a significant departure from the model of [4] and the resulting problem become quite non-trivial. However, it is a needed change and we have suggested novel and intuitive approximation scheme for solving the resulting problem.

4. Like in [4], we use the complex space $\mathbb{C}^d$ over the field of reals to embed binary predicates. Under the field of reals, the space $\mathbb{C}^d$ become isomorphic to $\mathbb{R}^{2d}$. We denote any binary predicate, say $r_k$ by its orthogonal projection matrix $\mathbf{P}_k$, and any entity pair $(e_i, e_j)$ by the vector $x_{ij} = [x_i, x_j] \in \mathbb{R}^{2d}$, where $x_i, x_j \in \mathbb{R}^d$ are the representation of individual entities $e_i, e_j$, respectively.

Like our main paper's IQE model for the unary predicate, the extension of IQE model for binary predicate case would result in the following optimization problem.

$$\underset{\{x_i\}_{i=1}^n, \{\mathbf{P}_k\}_{k=1}^m}{\text{Minimize}} \quad \sum_{i=1}^{n}\sum_{j=1}^{n}\sum_{k=1}^{m} \|\mathbf{Q}_k x_{ij}\|^2 \mathbb{1}_k(ij) + \lambda \|\mathbf{P}_k x_{ij}\|^2 \overline{\mathbb{1}}_k(ij)$$

$$= \sum_{i=1}^{n}\sum_{j=1}^{n} x_{ij}^{\top} \mathbf{R}(ij) x_{ij}$$

subject to
$$x_{ij} = [x_i, x_j]^{\top}; \; \forall i, j$$
$$\|x_i\|^2 = 1/2; \; \forall i$$
$$\mathbf{P}_k = \mathbf{V}_k \mathbf{D}_k \mathbf{V}_k^{\top}; \; \forall k$$
$$\mathbf{D}_k = \text{diag}(\{0, 1\}) \; \forall k$$
$$\mathbf{V}_k \mathbf{V}_k^{\top} = \mathbf{V}_k^{\top} \mathbf{V}_k = \mathbf{I} \; \forall k$$
$$\text{tr}(\mathbf{D}_k) \geq r; \; \forall k$$

where $\mathbf{Q}_k = (\mathbf{I} - \mathbf{P}_k)$ and the last four constraints together enforce the matrix $\mathbf{P}_k$ to be a projection matrix in $\mathbb{R}^{2d}$ space (not necessarily axis-parallel) of rank at least $r$. The first two constraints ensures

that $x_{ij}$ is a unit length vector in the space $\mathbb{R}^{2d}$. Like unary case, we propose an alternating scheme to solve the above problem as follows. Note, unlike unary predicate case, we don't have Problem 2 here because we have skipped modeling initial feature vectors. Therefore, we only talk about Problem 1 and Problem 3 in this case. Involving Problem 2 is straightforward.

---

**Algorithm 1:** Alternating Minimization Scheme for IQE Problem for Binary Predicates

---

Pick appropriate values for the hyperparameters $d, \lambda, r$ ;
Given a KB, construct the indicator function $\mathbb{1}_k(ij)$ and initialize $x_i$ randomly;
**while** *(Solution does not converge)* **do**

$\quad$ Clamp variables $\{\mathbf{P}_k\}_{k=1}^m$ and solve the resulting problem for $\{x_{ij}\}_{i,j=1}^n$. **[call Problem 1]**;
$\quad$ Clamp variables $\{x_{ij}\}_{i,j=1}^n$ and solve the IQE problem over $\{\mathbf{P}_k\}_{k=1}^m$ **[call Problem 3]**;

---

### 4.1 Solution for Problem 1 (Binary Predicate Case):

Note, if we clamp $\{\mathbf{P}_k\}_{k=1}^m$ satisfying the last four constraints of the formulation (10) then, the resulting Problem 1 can be written as follows:

$$\underset{\{x_i\}_{i=1}^n}{\text{Minimize}} f(\{x_i\}_{i=1}^n) = \sum_{i=1}^n \sum_{j=1}^n \sum_{k=1}^m (x_{ij}^\top \mathbf{Q}_k x_{ij}) \mathbb{1}_k(ij) + \lambda(x_{ij}^\top \mathbf{P}_k x_{ij}) \overline{\mathbb{1}}_k(ij)$$

$$= \sum_{i=1}^n \sum_{j=1}^n x_{ij}^\top \mathbf{R}(ij) x_{ij}$$

$$\text{subject to} \qquad x_{ij} = [x_i, x_j]^\top; \ \forall i, j$$
$$\|x_i\|^2 = 1/2; \ \forall i$$

where, we have made use of the fact that $\mathbf{Q}_k^2 = \mathbf{Q}_k$ and $\mathbf{P}_k^2 = \mathbf{P}_k$ because they are projection matrices. For each $(i, j)$ pair, we define the following matrix.

$$\mathbf{R}(ij) = \sum_{k=1}^m \left( \mathbf{Q}_k(ij) \mathbb{1}_k(ij) + \lambda \mathbf{P}_k(ij) \overline{\mathbb{1}}_k(ij) \right) \tag{10}$$

$$= \begin{bmatrix} \mathbf{R}^s(ij) & \mathbf{R}^c(ij) \\ \mathbf{R}^c(ij)^\top & \mathbf{R}^o(ij) \end{bmatrix} \tag{11}$$

where, due to symmetric PSD nature of projection matrices $\mathbf{P}_k$ and $\mathbf{Q}_k$, we have following holding true.

- Eigen decomposition of the matrix $\mathbf{P}_k$ is given by $\mathbf{V}_k \mathbf{D}_k \mathbf{V}_k^\top$.
- Each of the block matrix is of size $d$-by-$d$
- Block matrices $\mathbf{R}^s(ij)$, $\mathbf{R}^o(ij)$ are symmetric PSD for all $i, j$.

We have placed superscripts on these block matrices to indicate their position as subject $(s)$, object $(o)$, and cross-term $(c)$. In light of this definition, we can rewrite the above formulation as follows.

$$\underset{\{x_i\}_{i=1}^n}{\text{Minimize}} \quad \sum_{i=1}^n \sum_{j=1}^n x_i^\top \mathbf{R}^s(ij) x_i + x_j^\top \mathbf{R}^o(ij) x_j + x_i^\top \left( \mathbf{R}^c(ij) + \mathbf{R}^c(ij)^\top \right) x_j$$
$$\text{subject to} \quad \|x_i\|^2 = 1/2; \ \forall i \tag{12}$$

Note, matrices $\mathbf{R}^s(ij), \mathbf{R}^o(ij), \mathbf{R}^c(ij)$ are data matrices for the Problem 1 and hence they are constant. In order to solve above problem, we define following quantities

$$\varphi(x_i) = \sum_{j=1} x_i^\top \left( \mathbf{R}^s(ij) + \mathbf{R}^o(ij) \right) x_i + 2 \sum_{j=1}^n x_i^\top \left( \mathbf{R}^c(ij) + \mathbf{R}^c(ij)^\top \right) x_j \tag{13}$$

In light of this definition, we can express the objective function $f(\{x_i\}_{i=1}^n)$ as follows.

$$f(\{x_i\}_{i=1}^n) = \sum_{i=1}^n \varphi(x_i) - \sum_{i=1}^n \sum_{j=1}^n x_i^\top \left( \mathbf{R}^c(ij) + \mathbf{R}^c(ij)^\top \right) x_j \tag{14}$$

Further, we observe that

$$\frac{\partial \varphi(x_i)}{\partial x_i} = 2 \sum_{j=1}^n (\mathbf{R}^s(ij) + \mathbf{R}^o(ij))x_i + 2 \sum_{j=1}^n \left( \mathbf{R}^c(ij) + \mathbf{R}^c(ij)^\top \right) x_j \tag{15}$$

$$\implies \varphi(x_i) = \frac{1}{2} \left( x_i^\top \frac{\partial \varphi(x_i)}{\partial x_i} \right) + \sum_{j=1}^n x_i^\top \left( \mathbf{R}^c(ij) + \mathbf{R}^c(ij)^\top \right) x_j \tag{16}$$

Now, we write the Lagrangian of the Problem (12) as follows, where $\mu_i \in \mathbb{R}$ are the dual variables.

$$L\left(\{x_i\}_{i=1}^n, \{\mu_i\}_{i=1}^n\right) = \sum_{i=1}^n \varphi(x_i) - \sum_{i=1}^n \sum_{j=1}^n x_i^\top \left( \mathbf{R}^c(ij) + \mathbf{R}^c(ij)^\top \right) x_j + \sum_{i=1}^n \mu_i \left( x_i^\top x_i - \frac{1}{2} \right) \tag{17}$$

Observe, for each $i$, we must have

$$\mu_i \geq -\lambda_i \tag{18}$$

where $\lambda_i$ is the smallest eigenvalue of the matrix $\sum_{j=1}^n (\mathbf{R}^s(ij) + \mathbf{R}^o(ij))$. This is because, otherwise the Lagrange function would become unbounded from below and the value of Lagrange could be pushed to $-\infty$. Keeping this constraint on dual variables in mind, we now take the partial derivative of the Lagrange function with respect to primal variables and set them to zero. This yields the following:

$$\frac{\partial L(\{x_i\}_{i=1}^n)}{\partial x_i} = \frac{\partial \varphi(x_i)}{\partial x_i} - 2 \sum_{j=1}^n \left( \mathbf{R}^c(ij) + \mathbf{R}^c(ij)^\top \right) x_j + 2\mu_i x_i = 0 \tag{19}$$

$$\implies \left. \frac{\partial \varphi(x_i)}{\partial x_i} \right|_{x_i = x_i^*} = 2 \sum_{j=1}^n \left( \mathbf{R}^c(ij) + \mathbf{R}^c(ij)^\top \right) x_j - 2\mu_i x_i^*. \tag{20}$$

Substituting the value of (20) into (16), we get the following value for function $\varphi(x_i)$ at the point that minimizes Lagrangian:

$$\varphi(x_i^*) = 2 \sum_{j=1}^n x_i^{*\top} \left( \mathbf{R}^c(ij) + \mathbf{R}^c(ij)^\top \right) x_j^* - \mu_i x_i^{*\top} x_i^*. \tag{21}$$

Substituting the value of $\varphi(x_i^*)$ from Equation (13) into the above equation gives us the following relation:

$$x_i^{*\top} \left( \sum_{j=1}^n \mathbf{R}^s(ij) + \mathbf{R}^o(ij) + (\mu_i \mathbf{I}) \right) x_i^* = 0. \tag{22}$$

From the above characteristic equation about the primal optimal solution, and the facts that $\mathbf{R}^s(ij), \mathbf{R}^o(ij)$ are PSD matrices, we must have $\mu_i = -\lambda_i$, where $\lambda_i$ is the smallest eigenvalue of the matrix $\sum_{j=1}^n (\mathbf{R}^s(ij) + \mathbf{R}^o(ij))$. Also, we can choose $x_i^*$ to be the smallest eigenvector of the matrix $\left( \sum_{j=1}^n \mathbf{R}^s(ij) + \mathbf{R}^o(ij) \right)$ with length scaling of $1/\sqrt{2}$. Note, the primal optimal objective function value, therefore, would become as follows:

$$f(\{x_i^*\}_{i=1}^n) = \sum_{i=1}^n \left( \sum_{j=1}^n x_i^{*\top} \left( \mathbf{R}^c(ij) + \mathbf{R}^c(ij)^\top \right) x_j^* - \mu_i x_i^{*\top} x_i^* \right). \tag{23}$$

Although, the dual variables' values gets determined by looking at the above equation itself, let's write dual problem for the sake of completeness. Substituting the value of $\varphi(x_i^*)$ from Equation (21), for all $i$, into the Lagrangian function (17), we get the minimum value of the Lagrangian and

that would result in the following Lagrangian dual problem having the Lagrangian dual function $g(\{\mu_i\}_{i=1}^n)$.

$$\underset{\{\mu_i\}_{i=1}^n}{\text{maximize}} \quad g(\{\mu_i\}_{i=1}^n) = \sum_{i=1}^n \left( \sum_{j=1}^n x_i^{*\top} \left( \mathbf{R}^c(ij) + \mathbf{R}^c(ij)^\top \right) x_j^* - \frac{\mu_i}{2} \right) \tag{24}$$

$$\text{subject to} \quad \mu_i \geq -\lambda_i; \forall i \in [n].$$

We can see that by setting the dual variable value as $\mu_i^* = -\lambda_i$, the dual objective function value matches with the optimal primal objective function value ($x_i^{*\top} x_i = 1/2$). Therefore, we can conclude from this route also that the optimal value of the dual variable $\mu_i^*$ must be equal to $-\lambda_i$.

### 4.2 Solution for Problem 3 (Binary Predicate Case):

Note, if we clamp $\{x_{ij}\}_{i,j=1}^n$ satisfying the first two constraints of the IQE formulation, the resulting problem can be written as follows.

$$\underset{\{\mathbf{P}_k\}_{k=1}^m}{\text{Minimize}} \quad \sum_{i=1}^n \sum_{j=1}^n \sum_{k=1}^m \|\mathbf{Q}_k x_{ij}\|^2 \mathbb{1}_k(ij) + \lambda \|\mathbf{P}_k x_{ij}\|^2 \overline{\mathbb{1}}_k(ij)$$

$$\text{subject to} \quad \mathbf{P}_k = \mathbf{V}_k \mathbf{D}_k \mathbf{V}_k^\top; \ \forall k$$
$$\mathbf{D}_k = \text{diag}(\{0, 1\}) \ \forall k \tag{25}$$
$$\mathbf{V}_k \mathbf{V}_k^\top = \mathbf{V}_k^\top \mathbf{V}_k = \mathbf{I} \ \forall k$$
$$\text{tr}(\mathbf{D}_k) \geq r; \ \forall k$$

Recall that $\mathbf{Q}_k = \mathbf{I} - \mathbf{P}_k$ and the constraints basically force the matrix $\mathbf{P}_k$ to be a valid orthogonal projection matrix. In light of this, we can rewrite the objective function of (25) as follows.

$$f(\{\mathbf{P}_k\}_{k=1}^m) = \sum_{i=1}^n \sum_{j=1}^n \sum_{k=1}^m (x_{ij}^\top (\mathbf{I} - \mathbf{P}_k) x_{ij}) \mathbb{1}_k(ij) + \lambda (x_{ij}^\top \mathbf{P}_k x_{ij}) \overline{\mathbb{1}}_k(ij) \tag{26}$$

By ignoring the constant term, the function can be written as follows.

$$f(\{\mathbf{P}_k\}_{k=1}^m) = \sum_{i=1}^n \sum_{j=1}^n \sum_{k=1}^m -(x_{ij}^\top \mathbf{P}_k x_{ij}) \mathbb{1}_k(ij) + \lambda (x_{ij}^\top \mathbf{P}_k x_{ij}) \overline{\mathbb{1}}_k(ij)$$

$$= \sum_{i=1}^n \sum_{j=1}^n \sum_{k=1}^m -\text{tr}(x_{ij}^\top \mathbf{P}_k x_{ij}) \mathbb{1}_k(ij) + \lambda \ \text{tr}(x_{ij}^\top \mathbf{P}_k x_{ij}) \overline{\mathbb{1}}_k(ij) \tag{27}$$

$$= \sum_{i=1}^n \sum_{j=1}^n \sum_{k=1}^m -\text{tr}(\mathbf{P}_k x_{ij} x_{ij}^\top) \mathbb{1}_k(ij) + \lambda \ \text{tr}(\mathbf{P}_k x_{ij} x_{ij}^\top) \overline{\mathbb{1}}_k(ij) \tag{28}$$

$$= \sum_{i=1}^n \sum_{j=1}^n \sum_{k=1}^m \text{tr}\left[ \mathbf{P}_k \left( -x_{ij} x_{ij}^\top \mathbb{1}_k(ij) + \lambda \ x_{ij} x_{ij}^\top \overline{\mathbb{1}}_k(ij) \right) \right] \tag{29}$$

It is clear from above function that we can separate the objective function into $k$ and Problem 3 can be solved independently for each $k$. For a fixed $k$, the above objective function becomes

$$f(\mathbf{P}_k) = \sum_{i=1}^n \sum_{j=1}^n \text{tr}\left[ \mathbf{P}_k \left( -x_{ij} x_{ij}^\top \mathbb{1}_k(ij) + \lambda \ x_{ij} x_{ij}^\top \overline{\mathbb{1}}_k(ij) \right) \right] \tag{30}$$

$$= \text{tr}\left[ \mathbf{P}_k \sum_{i=1}^n \sum_{j=1}^n \left( -x_{ij} x_{ij}^\top \mathbb{1}_k(ij) + \lambda \ x_{ij} x_{ij}^\top \overline{\mathbb{1}}_k(ij) \right) \right] \tag{31}$$

$$= \text{tr}\left[ \mathbf{P}_k \mathbf{X}_k \right] \tag{32}$$

where, $\mathbf{X}_k = \sum_{i=1}^n \sum_{j=1}^n \left( -x_{ij} x_{ij}^\top \mathbb{1}_k(ij) + \lambda \ x_{ij} x_{ij}^\top \overline{\mathbb{1}}_k(ij) \right)$ and is a constant. In light of the above reformulation of the objective function, the solution of problem 3 would be as follows.

1. If we don't have rank constraints the above expression is minimized when $\mathbf{P}_k$ is chosen to be the projector onto the negative eigenspace of the matrix $\mathbf{X}_k$.

2. With rank constraint, we will need to take projection onto the smallest $s$ eigenvectors of the matrix $\mathbf{X}_k$, where $s = \max\{r, d_-\}$ where $r$ is the minimum rank and $d_-$ is the dimensionality of negative eigenspace of matrix $\mathbf{X}_k$.

## 5 Solution for Problem 1 (Unary Predicate Case): Optimizing over $x_i$

Observe, when $\mathbf{W}, \mathbf{P}_1, \mathbf{P}_2, \ldots \mathbf{P}_m$ are clamped to the values that satisfy constraints (5) and (6) of the main paper, the objective function (3) given in the main paper becomes convex quadratic in $x_i$'s. Furthermore, equality constraints are also quadratic in $x_i$'s. The resulting problem is known as Quadratically Constrained Quadratic Program (QCQP). Observe, such a QCQP problem is separable in the variables $x_1, \ldots, x_n$. Therefore, we can solve this QCQP problem by solving a separate problem for each $x_i$. Ignoring the constant term, the Problem 1 for an $x_i$ is given as follows.

$$\text{Minimize} \quad x_i^T \mathbf{R}_i x_i - 2 x_i^T c_i, \tag{33}$$

$$\text{subject to} \quad \|x_i\|^2 = 1, \tag{34}$$

$$\text{where} \quad \mathbf{R}_i = \alpha \mathbf{I}_d + \sum_{j=1}^m \mathbf{Q}_j \mathbb{1}_j + \lambda \mathbf{P}_j \bar{\mathbb{1}}_j \text{ and } c_i = \mathbf{W} f_i, \tag{35}$$

where $\mathbf{I}_d$ is a $d$-by-$d$ identity matrix. Let $L$ be a Lagrangian function with the Lagrange multiplier $\mu$ corresponding to the equality constraint,

$$L(x, \mu) \quad = \quad x_i^T \left( \mathbf{R}_i - \mu \mathbf{I} \right) x_i - 2 x_i^T c_i + \mu.$$

In order to minimize the Lagrangian with respect to $x_i$, it should be bounded from below. The above quadratic function is bounded below if and only if the Hessian $(\mathbf{R}_i - \mu \mathbf{I})$ is positive definite. The stationary values of the Lagrangian function gives,

$$\left( \mathbf{R}_i - \mu \mathbf{I} \right) x_i = c_i. \tag{36}$$

Using stationary condition, the Lagrangian dual function is

$$g(\mu) \quad = \quad \inf_{x_i} \left( x_i^T \left( \mathbf{R}_i - \mu \mathbf{I} \right) x_i - 2 x_i^T c_i \right) + \mu = -c_i^T \left( \mathbf{R}_i - \mu \mathbf{I} \right)^{-1} c_i + \mu. \tag{37}$$

Therefore, the Lagrangian dual problem is

$$\text{Maximize} \quad g(\mu)$$

$$\text{such that} \quad \left( \mathbf{R}_i - \mu \mathbf{I} \right) > 0. \tag{38}$$

Note that the dual constraint is satisfied if and only if $\mu$ is less than the smallest eigenvalues (say $\lambda_1$) of $\mathbf{R}_i$ i.e. $\mu < \lambda_{min}(\mathbf{R}_i)$. Noting that the $\mathbf{R}_i$ in our case is a diagonal matrix as the projection matrices $\mathbf{P}_j, \mathbf{Q}_j$'s are diagonal. The stationary value of the Lagrangian dual function gives rise to the following secular equation [9]

$$\sum_{j=1}^d \frac{c_{ij}^2}{(\lambda_j - \mu)^2} = 1 \text{ and } \mu < \lambda_1, \tag{39}$$

where, $c_{ij}$ is the $j^{th}$ component of the vector $c_i$. The LHS of the secular equation (39) is a monotonically increasing function of $\mu$ taking value in the range of $(0, +\infty)$ as we move $\mu$ in the interval $(-\infty, \lambda_1)$. Therefore, it must have one unique solution in the interval $(-\infty, \lambda_1)$. We obtained $\mu$ by solving (39) using *bisection method* [10].

## 6 Solution for Problem 3 (Unary Predicate Case): Optimizing over $\mathbf{P}_j$

Here, we consider the problem of optimizing over the subspaces when $x_1, \ldots, x_n$ and $\mathbf{W}$ are clamped to their current estimates. Since all the projection matrices $\mathbf{P}_j$'s commute, they are simultaneously diagonalizable via a common orthogonal matrix (due to Theorem 8 given in Section 1 of the supplementary material). Furthermore, because IQE is rotationally invariant, we can assume, without

loss of generality, the projection matrices to be diagonal. We, therefore, take each projection matrix $\mathbf{P}_j$ to be of the form $\text{diag}(y_{j,1}, ..., y_{j,d})$ where each $y_{j,k} \in \{0, 1\}$.

In what follows, we start analyzing the simpler version of Problem 3 for unary predicates, where we ignore the pairwise orthogonality term (i.e., last term) in the objective function (3), given in the main paper, as well as the rank constraint (6) of the main paper. Later, we will show how to incorporate rank constraint in Section 6.1 and to incorporate orthogonality term in Section 6.2. We will also discuss some heuristics to incorporate both of them together in Section 6.3.

The loss function without orthogonality term is given by the first term of the Equation (12) of the main paper. Note, this loss function is separable in $j$, and hence we can separately minimize the following problem for each $j$.

$$\text{Minimize} \qquad \sum_{k=1}^{d} y_{j,k} \phi_{j,k} \qquad (40)$$

$$\text{where,} \quad \phi_{j,k} \overset{def}{=} \lambda \sum_{i \notin S_j} x_{i,k}^2 - \sum_{i \in S_j} x_{i,k}^2 \qquad (41)$$

refers as the *potential function*. The objective function (40) is also separable in $k$, therefore it boils down to minimizing $y_{j,k} \phi_{j,k}$ for each $j$ and $k$. Therefore, depending upon the value of the potential function $\phi_{j,k}$ the following values of $y_{j,k}$ minimizes the term $y_{j,k} \phi_{j,k}$

$$y_{j,k} = \begin{cases} 1 & \text{if } \phi_{j,k} < 0 \text{ (i.e., } \sum_{i \in S_j} x_{i,k}^2 > \lambda \sum_{i \notin S_j} x_{i,k}^2 ) \\ 0 & \text{otherwise.} \end{cases} \qquad (42)$$

## 6.1 Adding Rank constraint

We now consider optimizing (40) under the constraints (6) of the main paper that the dimension of each subspace must be at least $r$. This is equivalent to constraining each $\mathbf{P}_j$ to have a rank at least $r$. Given the diagonal form of $\mathbf{P}_j$s, and dropping the constant term from the Problem 3 in the main paper, the problem now reduces to minimizing the objective (40)

$$\text{subject to} \qquad \sum_{k}^{d} y_{j,k} \geq r \;\; \forall j \qquad (43)$$
$$y_{j,k} \in \{0, 1\}.$$

In order to minimize this problem under the rank constraint (43), we consider two separate cases:

Case I: When at least $r$ of the $\phi_{j,k}$'s are $\leq 0$ or equivalently $|\{k : \phi_{j,k} \leq 0\}| \geq r$, the solution (42) to the previous problem also satisfies the new constraints since at least $r$ of the $y_{j,k}$'s are 1.

Case II: This is the case when we have $|\{k : \phi_{j,k} \leq 0\}| < r$. In this case, we consider a permutation of the indices from 1 to $d$ as $k_1, ..., k_d$ such that $\phi_{j,k_1} \leq \phi_{j,k_2}, ... \leq \phi_{j,k_d}$. The minimum value of the objective (40) can be achieved while maintaining the constraint that sum of $y_{j,k}$ is at least $r$ by choosing the first $r$ $y_{j,k_l}$'s to be 1 and remaining to be 0.

## 6.2 Adding Orthogonality Term

We now solve Problem 3 of the main paper without rank constraint but requiring that the projection subspaces are roughly orthogonal to each other. That is, by including the second term of Equation (12) in the main paper but ignoring the rank constraint (43). We avoid imposing orthogonality as hard constraints since some of the concepts can have an overlapping set of entities. Due to the diagonal form of $\mathbf{P}_j$'s, and dropping constant term from Problem 3, the problem reduces to

$$\text{minimize} \sum_{k=1}^{d} \left( \sum_{j=1}^{m} y_{j,k} \phi_{j,k} + \gamma \sum_{j' > j} y_{j,k} y_{j',k} \right), \qquad (44)$$
$$\text{subject to} \;\; y_{j,k} \in \{0, 1\}.$$

Here, we observe that although the objective function is not separable in $j$ but is separable in $k$. Each constraint is also separable in $k$. For each $k$, we need to minimize

$$\sum_{j=1}^{m} y_{j,k}\phi_{j,k} + \gamma\binom{n_k}{2},\tag{45}$$

where $n_k = |\{j : y_{j,k} = 1\}|$. With the above formulation of the objective, we can draw 2 observations:

1. If we were upfront told that the value of $n_k$ in the optimal solution is, say $t$. Then, we can infer that only those $y_{j,k}$'s would be having values as one which corresponds to the smallest $t$ values of $\phi_{j,k}$. It is because for any solution having $n_k = t$, we can always create a solution of the same or a lower objective value by setting $t$ of those $y_{j,k}$'s as one which has lowest $t$ values of $\phi_{j,k}$'s.

2. Suppose we have chosen $(\ell - 1)$ smallest entries and we denote the $\ell^{th}$ smallest entry by $\phi_{j_\ell,k}$. Then, additional contribution of adding the $\ell^{th}$ smallest entry to the solution is $\phi_{j_\ell,k} + \gamma(\binom{\ell}{2} - \binom{\ell-1}{2})$ which is also equal to $\phi_{j_t,k} + \gamma(\ell - 1)$. This increment also increases for each successive $\ell$. Thus, once it becomes positive it remains so from then on.

From the above two observations, we see that it suffices to sort all the $\phi_{j,k}$'s in increasing order and then greedily keep assigning $y_{j,k} = 1$ until the objective function value continues to decrease. The steps are given in Algorithm 2.

---

**Algorithm 2:** Solution of Problem 3 for fixed $k$ when Ignoring Rank Constraints

---
Initialize $y_{j,k} = 0 \;\; \forall j$;
Sort indices $j$ in the increasing order of $\phi_{j,k}$ and call them as $j_1, \ldots, j_m$;
**for** $\ell \leftarrow 1$ **to** $m$ **do**
    **if** $\phi_{j_\ell,k} + \gamma(\ell - 1) \leq 0$ **then**
        $y_{j_\ell,k} = 1$;
    **else**
        Break;
    **end**
**end**

---

## 6.3 Joint Optimization with Orthogonality Term + Rank Constraint

In this case, we minimize (44) subject to rank and binary constraints

$$\sum_{k=1}^{d} y_{j,k} \geq r \;\; \text{and} \;\; y_{j,k} \in \{0,1\}.$$

It is difficult to solve Problem 3 efficiently when both orthogonality terms and rank constraints are considered together. This is because the objective (44) is separable in $k$, but the rank constraint is not separable in $k$. For this, we will instead apply some heuristics to solve it approximately. We first solve Problem 3 with rank constraint alone, as discussed in Section 6.1. Subsequently, we greedily drop some of the $y_{j,k}$'s, which help decrease the overall objective function, including the orthogonality term. This, however, must be done without compromising on the rank constraint. The heuristic is given as Algorithm 3.

## 6.4 An Alternate Heuristic

An alternative heuristic to solve Problem 3 with joint constraints of orthogonality and rank could be as follows. We first optimize with the orthogonality term but without the rank constraint. Subsequently, we greedily add some of the $y'_{j,k}s$ so as to be able to fulfill the rank constraint. The pseudo-code is

**Algorithm 3:** A Heuristics to Solve Problem 3 for Unary Predicate

---

Solve Problem 3 without orthogonality term as described in Section 6.1.;
$V = \{(j,k) : y_{j,k} = 1, \sum_{k'=1}^{d} y_{j,k'} > r, \text{ and } (\phi_{j,k} + \gamma(n_k - 1)) > 0\}$ ;
**while** $V \neq \emptyset$ **do**
  $(j^*, k^*) \leftarrow \text{argmax}_{(j,k) \in V} [\phi_{j,k} + \gamma(n_k - 1)]$;
  $y_{j^*,k^*} \leftarrow 0$;
  $n_{k^*} - = 1$;
  $V = \{(j,k) : y_{j,k} = 1, \sum_{k'=1}^{d} y_{j,k'} > r, \text{ and } (\phi_{j,k} + \gamma(n_k - 1)) > 0\}$;
**end**

---

given in Algorithm 4.

**Algorithm 4:** An Alternate Heuristics to Solve Problem 3 for Unary Predicate

---

Solve the Problem 2 without rank constraint.;
$V = \{(j,k) : y_{j,k} = 0 \text{ and } \sum_{k'=1}^{d} y_{j,k'} < r\}$
**while** $V \neq \emptyset$ **do**
  $(j^*, k^*) \leftarrow \text{argmin}_{(j,k) \in V} [\phi_{j,k} + \gamma n_k]$
  $y_{j^*,k^*} \leftarrow 1$
  $n_{k^*} + = 1$
  $V = \{(j,k) : y_{j,k} = 0 \text{ and } \sum_{k'=1}^{d} y_{j,k'} < r\}$
**end**

---

At each iteration we choose the solution with minimum cost amongst those produced by Algorithms 3 and 4.

## 7 Experiment

### 7.1 Hierarchy of FIGER dataset

The Figure 1 depicts the *fine-grained entity type hierarchy* present in the FIGER dataset. Here, each cell corresponds to one parent node and all its children node. The label of the parent node is always denoted in red bold colored text whereas the labels of its children nodes are denoted in black colored text. The second last cell corresponds to the leaf nodes which are directly connected to the root of the type hierarchy. The last cell just depicts that all the internal nodes are indeed children of the root node, justifying two levels of the hierarchy.

This hierarchy consists of 127 different entity types arranged in two level of the hierarchy. The leaf nodes in this hierarchy are 106 and the non-leaf nodes are 21. From the original hierarchy given in the FIGER dataset [11], we have made a few minor modifications for the sake of maintaining consistency.

- Replaced `computer` with `computer_science`
- Replaced `religion/religion` with `religion`
- Replaced `government/government` with `government/administration`

| | | | | | |
|---|---|---|---|---|---|
| **art**<br>film | **broadcast**<br>tv_channel | **building**<br>airport<br>dam<br>hospital<br>hotel<br>library<br>power_station<br>restaurant<br>sports_facility<br>theater | **computer_science**<br>algorithm<br>programming_language | **education**<br>department<br>educational_degree | **event**<br>attack<br>election<br>military_conflict<br>natural_disaster<br>protest<br>sports_event<br>terrorist_attack |
| **finance**<br>currency<br>stock_exchange | **geography**<br>glacier<br>island<br>mountain | **government**<br>administration<br>political_party | **internet**<br>website | **livingthing**<br>animal | **location**<br>body_of_water<br>bridge<br>cemetery<br>city<br>country<br>county<br>province |
| **medicine**<br>drug<br>medical_treatmen<br>symptom | **metropolitan_transit**<br>transit_line | **organization**<br>Airline<br>company<br>educational_institution<br>fraternity_sororit<br>sports_league<br>sports_team<br>terrorist_organization | **people**<br>ethnicity | **person**<br>actor<br>architect<br>artist<br>athlete<br>author<br>coach<br>director<br>doctor<br>engineer<br>monarch<br>musician<br>politician<br>religious_leader<br>soldier<br>terrorist | **product**<br>airplane<br>camera<br>car<br>computer<br>engine_device<br>Instrument<br>mobile_phone<br>ship<br>spacecraft<br>weapon |
| **rail**<br>railway | **transportation**<br>road | **visual_art**<br>color | **<root node>**<br>astral_body<br>award<br>biology<br>body_part<br>broadcast_network<br>broadcast_program<br>chemistry<br>disease<br>food<br>game<br>god<br>government_agency<br>language<br>law<br>living_thing<br>military<br>music<br>news_agency<br>newspaper<br>park<br>play<br>religion<br>software<br>time<br>title<br>train<br>transit<br>written_work | **<root node>**<br>**art**<br>**broadcast**<br>**building**<br>**computer_science**<br>**education**<br>**event**<br>**finance**<br>**geography**<br>**government**<br>**internet**<br>**livingthing**<br>**location**<br>**medicine**<br>**metropolitan_transit**<br>**organization**<br>**people**<br>**person**<br>**product**<br>**rail**<br>**transportation**<br>**visual_art** | |

Figure 1: Fine-grained Entity Type Hierarchy in FIGER dataset.