[Reviews · NeurIPS 2020]

Review 1

Summary and Contributions: The paper extends a technique for geometrical embedding of logical hierarchies, known as 'quantum embedding' introduced by [Garg 2019]. Quantum Embedding encodes entities as vectors in a high-dimensional space, and concepts as linear subspaces, where the probability that an entity belongs to a subspace is proportional to the squared norm of it's projection. This principle mimics the principle of measurement in quantum mechanics, which leads to the name of the procedure (Note that this is a completely classical algorithm and does not have any relationship to quantum computing). Furthermore, it is desired that the subspaces obey certain relationships that would encode their logical structure, for example the negation of a subspace is it's orthogonal complement, the AND is it's intersection and the OR is it's vector sum. The main question that is addressed in the paper is how one can construct embeddings that approximately satisfy these constraints. In the original work of [Garg et. al 2019] the desired properties are encoded in a non-convex loss function that is minimized by SGD. The current paper reformulates this optimization problem and claims 2 main benefits over previous work. 1) The procedure is inductive and provides and allows the embedding of a new test point to be computed incrementally. 2) An alternating minimization method is developed for optimizing the loss function, that is empirically substantially faster than the scheme developed in [Garg 2019]. The scheme is empirically tested on a task for Fine-Grained Entity Type Classification and achieves state of the art accuracy on benchmark test sets, and close to the state of the art in F1 and F2, while running much faster than the previous procedure of Garg et. al.

Strengths: Quantum Embedding seems like a promising approach and thus improvements to the pipeline for computing it are quite valuable, especially given the non-convex regime of optimization. The ability to incrementally compute test embeddings is also a massive improvement at inference time.

Weaknesses: The state of the art results accuracy presented seen to be a feature more of the quantum embedding approach than the exact improvements presented here. It would be nice to have an accuracy comparison with the original method of Garg et. al. (EDIT: In the author feedback it is pointed out that such a comparison does exist in the paper. I have increased my score to reflect this.) The observations made regarding the geometry of the computed embeddings are not very surprising or informative. Rotational invariance of the embedding is almost necessitated since every quantity of interest in the embeddings is rotationally invariant. The discussion at the end of Section 3, mostly contributes to understanding the role of various optimization terms and hyperparameters rather than the embedding itself.

Correctness: I did not find any issues with correctness.

Clarity: The paper is mostly clear but can be a little difficult to parse for someone new to the fields of Knowledge Representation, and the associated tasks in machine learning and NLP. A gentler introduction to these would have been appreciated.

Relation to Prior Work: Yes.

Reproducibility: Yes

Additional Feedback:


Review 2

Summary and Contributions: This submission is continued research on the use of quantum logic inspired embedding for knowledge representation (KR).  It provides two improvements over the prior-art [Garg et al, NeurIPS 19]: (1) allowing inductive reasoning of quantum embedding; (2) a faster training method (via empirical comparison). 

Strengths: The new inductive reasoning functionality is highly desirable for KR. The formulation of the Inductive Quantum Embedding problem is nice and intuitive. The empirical evaluation part is also well designed.

Weaknesses: Although one can intuitively understand a few nice properties of the Inductive Quantum Embedding (IQE) problem, it is not clear whether the current formulation is the only choice. Since the evaluation of this optimization formulation, especially its efficiency, is fully empirical, it would be more convincing if one can show other simple alternatives of IQE is less promising.

Correctness: The theoretical part looks sound although the reviewer hasn’t checked all the details. The empirical part is solid.

Clarity: Mostly well written.

Relation to Prior Work: It would be better if authors could elaborate more about the tasks implemented in [Garg et al 19]. The review had to read the original paper to see why inductive reasoning was not implemented in [Garg 19]. Post-Author-Response: thanks for promising to include more details about previous work.

Reproducibility: Yes

Additional Feedback:


Review 3

Summary and Contributions: The paper proposes two advancements over Quantum Embedding: making it inductive instead of transductive and making it faster. The authors show an application to fine-grained entity type classification. In this case, the authors compute the quantum embeddings of the entities in the knowledge base and then train a NN to predict the embeddings. This NN is then used at test time to produce the embedding of the test entity that is used to determine class membership. Differently from the original QE paper, they do not use SGD to compute the embeddings but an alternating optimization algorithm. The author show that their algorithm is 9 times faster than SGD. Application to the FIGER dataset shows that the approach can achieve the best accuracy, even though the second best F1.

Strengths: The authors tackle an interesting problem, that of embedding entities of knowledge bases in a way such that the logical structure of the kb is preserved. Quantum embedding is shown to be more accurate that for example Glove embedding (figure 3 and 4). The proposal makes it inductive permitting its use to classify unseen entities. Moreover, the proposal is faster than the previous algorithm.

Weaknesses: While the algorithm is shown to be faster than QE, the comparison does not take into account the time for training the NN. The performance on FIGER are second best for F1, even though the method is more generally applicable than the best one for F1.

Correctness: The theoretical claims seem sound, the main proofs are in supplementary material The proofs are quite involved, I could follow the general reasoning but sometimes not the details.

Clarity: The paper is well written, the concept are clearly explained.

Relation to Prior Work: Prior work seems to be taken correctly into account.

Reproducibility: Yes

Additional Feedback: Please report also on NN training time. --------------------- After reading the other reviews and the feedback, I believe the authors have satisfactorily addressed the comments so I will keep my score

[Author Response · NeurIPS 2020]

**All Reviewers:** We thank all the reviewers for your valuable time and insightful reviews. We also thank your kind
appreciation for our ideas and experimental setup to show inductive extension and speed up in the original quantum
embedding proposal. We have tried our best to clarify your questions. [A cryptic form of reviewer's question/comment
precedes our response.]

**Reviewer #2**:
[Accuracy comparison with the original method of Garg et al.] We believe we had given the accuracy numbers for
the original method of Garg et al. in Table 4 (first row) of the main paper. Here are the performance numbers for the
original method (Garg et. al.): **0.383 (accuracy), 0.420 (Macro F1), 0.347 (Micro F1)**. Our performance number on
the same task are as follows: **0.631(accuracy), 0.764 (Macro F1), 0.724 (Micro F1)**. Also, we would like to add, in
this paper, we are working with fine grained entity classification task which was not considered in the original paper.
[Rotational invariance of the embedding is almost necessitated.] We agree, prima facie, the observation regarding
rotational invariance property may not be very surprising. However, we explicitly called this out because of two reasons:
(i) it plays a crucial role in the formulation as well as the solution of the subproblem 3 (line # 180-184 in the main
paper), (ii) moreover; we felt this property might not strike to a reader's mind in an obvious manner while glancing
through the nonlinear and non-convex formulation of the QE problem.
[The discussion at the end of Section 3, mostly contributes to understanding the role of various optimization terms and
hyperparameters rather than the embedding itself.] Yes, you are right. However, we added this section, realizing that
such a discussion can tremendously help a reader understand and appreciate the formulation as well as the solution of
the subproblem 3, which otherwise may sound a bit intricate. We felt such a discussion could offer an intuitive and
geometric feel in reader's mind regarding why an axis is chosen for a specific subspace by our algorithm. Also, the
reader can convince oneself that the behavior of our proposed algorithm for subproblem 3 is indeed a natural way of
generating concept subspaces.
[A gentler introduction to Knowledge Representation and the associated tasks would have been appreciated.] Thanks
for pointing it out. We will certainly add a gentle introduction to these topics in the camera-ready version.

**Reviewer #3**
[Whether the current formulation is the only choice. It would be more convincing if one can show other simple
alternatives of IQE is less promising.] This is a very good suggestion. Although we had given a careful thought while
formulating the optimization problem (especially, in designing the rank constraints and the orthogonality constraints),
we will shed more light on this aspect in the final version of the paper. Just to add, in sections 6.1 and 6.2 of the
supplementary material, we have worked out two different simplified versions of the IQuE problem by relaxing either
of these two constraints (orthogonality and rank). During our experiments, as stated in line # 126-128 of the main paper,
we found that these simpler formulations result in degenerate solutions for the subproblem 3, where multiple concept
subspaces get collapsed to zero and thereby resulting in inferior quality embedding.
[ The review had to read the original paper to see why inductive reasoning was not implemented in [Garg 19].] We are
sorry for the inconvenience. Thanks for pointing it out. In the camera-ready version, we will certainly elaborate on the
task implemented (and their non-inductive nature) in the original paper of Garg et al.

**Reviewer #5**
[While the algorithm is shown to be faster than QE, the comparison does not take into account the time for training the
NN.] We want to highlight that NN comes after QE in our pipeline (as shown in Figure 2). The NN part learns the
mapping from the input sentence feature vector to the QE, irrespective of which method was used to generate the QE
(our method or the original method). Therefore, we felt the meaningful comparison would be to compare only in terms
of the time taken to generate QE (by the original method of Garg et al. versus our proposed method). However, for the
sake of completeness, we are providing below a table comparing two approaches including the training time of NN as
well. In this table, $t_{qe}$ and $i_{qe}$ denote *per iteration time* and *number of iterations*, respectively, taken during the training
of any QE method. The quantities $t_{nn}$ and $i_{nn}$ denote *average time per epoch* and *number of epochs* respectively, for
the training of NN. The average time $t_{nn}$ is approximately the same for both the methods. There are now two speedup
factors - one without including $T_{nn}$ and other with including $T_{nn}$.

| Method | $t_{qe}$(in sec.) | $i_{qe}$ | $T_{qe} = t_{qe} \times i_{qe}$ | $t_{nn}$ (in sec.) | $i_{nn}$ | $T_{nn} = t_{nn} \times i_{nn}$ | $T_{qe} + T_{nn}$ |
|---|---|---|---|---|---|---|---|
| Ours | 510.6 | 6 | 3063.6 | $\approx 1275$ | 6 | 7650 | 10713.6 |
| Garg et. al. 2019 | 27.8 | 1000 | 27800.0 | $\approx 1275$ | 6 | 7650 | 35450.0 |
| Speedup Factor | | | **9.07** | | | | **3.31** |

47
[The performance on FIGER are second best for F1, even though the method is more generally applicable than the best
one for F1.] Yes, you are absolutely right. Even though our method achieved second best F1 on fine-grained entity
task, unlike the best F1 method [i.e. *Attentive* (Shimaoka 2016)], our method is task agnostic and is more generally
applicable to many downstream tasks. Also, note that our method beats the *Attentive* method in terms of accuracy.

[Meta-Review · NeurIPS 2020]

The paper presents an extension of the quantum embeddings of (Gang et al., 2019) -- embeddings that allow for logical expressions to be evaluated. The main contributions are to allow for inductive learning of quantum embeddings, and the design of an algorithm that is significantly faster than the previous one. The experiments show promising results on a fine-grained classification task. The reviewers agreed that the paper presents a solid contribution and the rebuttal answered the reviewers concerns.